# In vivo armed macrophages curb liver metastasis through tumor-reactive T-cell rejuvenation

Marco Notaro [1,2], Maristella Borghetti[1,2], Chiara Bresesti [1], Giovanna Giacca[1,2], Thomas Kerzel [1], Carl Mirko Mercado [1], Stefano Beretta [3], Marco Monti [3], Ivan Merelli[3], Silvia Iaia[4], Marco Genua [5], Andrea Annoni [4], Tamara Canu[6], Patrizia Cristofori[7], Sara Degl'Innocenti[7], Francesca Sanvito [7,8], Paola Maria Vittoria Rancoita[9], Renato Ostuni [2,5], Silvia Gregori [4], Luigi Naldini [2,10] & Mario Leonardo Squadrito [1,2] ✉

Despite recent progress in cancer treatment, liver metastases persist as an unmet clinical need. Here, we show that arming liver and tumor-associated macrophages in vivo to co-express tumor antigens (TAs), IFNα, and IL-12 unleashes robust anti-tumor immune responses, leading to the regression of liver metastases. Mechanistically, in vivo armed macrophages expand tumor reactive CD8[+] T cells, which acquire features of progenitor exhausted T cells and kill cancer cells independently of CD4[+] T cell help. IFNα and IL-12 produced by armed macrophages reprogram antigen presenting cells and rewire cellular interactions, rescuing tumor reactive T cell functions. In vivo armed macrophages trigger anti-tumor immunity in distinct liver metastasis mouse models of colorectal cancer and melanoma, expressing either surrogate tumor antigens, naturally occurring neoantigens or tumor-associated antigens. Altogether, our findings support the translational potential of in vivo armed liver macrophages to expand and rejuvenate tumor reactive T cells for the treatment of liver metastases.

Despite remarkable progress in cancer treatment, including the latest advances in immunotherapy, liver metastases (LM) persist as an unmet clinical challenge. LM arising from distinct types of primary tumors, including immunotherapy-responsive colorectal carcinoma (CRC) and melanoma, poorly respond to current pharmacological treatments[1,2]. This observation might be attributed, at least in part, to the immunosuppressive characteristics of the liver, which, due to its close

connection with the digestive tract and its role in detoxification and blood filtration, has adapted to attenuate local immune reactions[3,4]. This condition favors the seeding and growing of LM and reduces the efficacy of therapies aimed at activating immune responses.

It is increasingly appreciated that LM evades anti-tumor immune responses by inhibiting tumor-specific T cells or by recruiting immunosuppressive cell types[5–8]. In this context, immunotherapeutic

[1]Vector Engineering and In vivo Tumor Targeting Unit, San Raffaele Telethon Institute for Gene Therapy, IRCCS San Raffaele Scientific Institute, Milan, Italy. [2]Vita-Salute San Raffaele University, Milan, Italy. [3]BioInformatics Core, San Raffaele Telethon Institute for Gene Therapy, IRCCS San Raffaele Scientific Institute, Milan, Italy. [4]Mechanisms of Peripheral Tolerance Unit and Immune Core, San Raffaele Telethon Institute for Gene Therapy, IRCCS San Raffaele Scientific Institute, Milan, Italy. [5]Genomics of the Innate Immune System Unit, San Raffaele Telethon Institute for Gene Therapy, IRCCS San Raffaele Scientific Institute, Milan, Italy. [6]Preclinical Imaging Facility, IRCCS San Raffaele Scientific Institute, Milan, Italy. [7]GLP Test Facility, San Raffaele Telethon Institute for Gene Therapy, IRCCS San Raffaele Scientific Institute, Milan, Italy. [8]Pathology Unit, IRCCS San Raffaele Scientific Institute, Milan, Italy. [9]CUSSB University Center for Statistics in the Biomedical Science, Vita-Salute San Raffaele University, Milan, Italy. [10]Targeted Cancer Gene Therapy Unit, San Raffaele Telethon Institute for Gene Therapy, IRCCS San Raffaele Scientific Institute, Milan, Italy. ✉e-mail: squadrito.mario@hsr.it

approaches might provide a solution since they are aimed at reinvigorating adaptive immune responses against the tumor. T cell-based strategies, including TCR-redirected T cells and chimeric antigen receptor (CAR) T cells, offer a promising approach by generating and expanding tumor-specific T cells ex vivo. However, these technologies suffer from the necessity of adopting specific TCR or CARs that recognize a selected antigen that is expressed on most cancer cells, but not on healthy tissues. Moreover, current T cell therapies necessitate ex vivo manipulation and patient conditioning, which results in delayed implementation, and present challenges due to the specialized expertize required for cell engineering[9]. Tumor vaccination overcomes some of the limitations of ex vivo T cell therapies by enabling the expansion of endogenous tumor-reactive T cells. After recent groundbreaking applications of RNA-based vaccines in the COVID-19 pandemic, there has been growing interest in the application of this technology for the development of innovative tumor vaccines[10,11]. However, tumor vaccines, as well as T cell therapies, suffer from the immunosuppressive signals within tumor, leading to suboptimal therapeutic responses[12,13]. Immune-activating cytokines may constitute a valuable resource to unleash adaptive immune responses. Individually, interferon-a (IFNα) and interleukin-12 (IL-12) have been employed in clinical and preclinical studies to treat distinct types of tumors, including LM, obtaining promising therapeutic results[14,15]. In vivo, gene therapy may constitute a valuable option to simultaneously expand tumor-specific T cells and reprogram their genetic program, overcoming some of the limitations associated with T cell therapies, tumor vaccines, or cytokines-based treatments alone.

Here, we describe a lentiviral vector (LV)-based gene therapy approach to enforce coordinated expression of tumor antigens (TAs), IFNα, and IL-12 selectively in liver and tumor-associated macrophages (TAMs) after single intravenous administration. Armed liver macrophages driving concurrent TA and cytokine expression inhibit tumor growth in distinct CRC and melanoma LM mouse models leading to complete response in most mice. Tumor inhibition is associated with a wide reprogramming of the tumor microenvironment (TME) as well as expansion, activation, and rejuvenation of TA reactive CD8+ T cells. Besides describing a promising therapeutic tool, our findings provide insights into the genes, pathways, and mechanisms that drive and promote immune activation and reprogramming in the metastatic liver.

## Results

### Coordinated IFNα and IL-12 expression rescues tumor-reactive T-cell activity

We first investigated whether liver macrophages expressing TAs enabled protective immunity against tumors. To this aim, we employed LVs incorporating a macrophage-specific promoter (i.e., *Mrc1*), along with microRNA target sites for miR-122-5p and miR-126-3p that inhibit off-target transgene expression in hepatocytes and liver sinusoidal endothelial cells. Previously, we showed that transgene expression of this LV platform is selective for liver macrophages, with minor expression in splenic MRC1+ macrophages. Of note, we showed no expression in other biological compartments such as blood, bone marrow, lung, sub-iliac lymph nodes, small intestine, or brain[15]. We included in this macrophage-specific cassette a surrogate TA, chicken ovalbumin (OVA). We employed a version of OVA fused to the CD74 invariant chain (IiOVA), which, upon processing in antigen presenting cells (APCs), is presented in both MHC-I and MHC-II complexes (Fig. 1A)[16,17]. As control, we generated a LV containing the same regulatory sequences of IiOVA LV, but lacking a coding sequence (Control, Fig. 1A). We injected Control LV at a single high dose ($10^8$ TU/mouse) and IiOVA LV at a low (i.e., $10^6$ TU/mouse), intermediate (mid, i.e., $10^7$ TU/mouse), or high dose (i.e., $10^8$ TU/mouse) intravenously (i.v.) in immunocompetent C57BL/6 mice (Fig. 1B). After 14 days from LV treatment, to model CRC-LM, we implanted intrahepatically a mixed population of CRC-like cell line, MC38, either untransduced or

transduced with an OVA LV, at a ratio of 1:9, respectively, hereon MC38.OVA. Treatment with a mid-dose of IiOVA LV resulted in a significant reduction in tumor growth, with 5/9 mice completely eradicating LM (Fig. 1C).

We observed expansion of H2kb/SIINFEKL tetramer+ CD8+ T cells, hereon indicated as OVA-specific CD8+ T cells, in the blood and liver of IiOVA-treated mice in a dose-dependent fashion (Fig. 1D, E). Previous studies have shown that exhausted T cells can be distinguished into distinct subsets, including terminally exhausted (TEX, EOMES+ PD1high) and progenitor exhausted (PEX, Tbet+ PD1int) CD8+ T cells. The former is associated with a dysfunctional CD8+ T-cell phenotype, whereas the latter is associated with greater polyfunctionality, longer persistence, and tumor control[18,19]. The level and persistence of antigen presentation play a key role in determining the fate of activated CD8+ T cells and it has been shown that high levels of TA presentation lead to defective CD8+ T cell activation[20,21]. Consistent with these studies, IiOVA delivery induced a dose-dependent accumulation of TEX OVA-specific CD8+ T cells, accompanied by a corresponding reduction in PEX OVA-specific cells (Fig. 1F and Supplementary Fig. 1A, B).

To further assess the effect of liver-macrophage-specific TA presentation on systemic immunity, we challenged mice injected with mid or high IiOVA dose with subcutaneous (s.c) MC38.OVA tumors. Delivery of IiOVA LV before tumor implantation reduced MC38.OVA s.c. tumor growth in mice treated with a mid dose of the LV but not with the high dose (Supplementary Fig. 2A). In agreement with this observation, MC38 clones expressing OVA were cleared in the tumors from the IiOVA mid dose, but not in the Control or IiOVA high dose groups (Supplementary Fig. 2B). We observed expansion of OVA-specific CD8+ T cells, in the liver in IiOVA mid and IiOVA high dose groups, but not in Control mice (Supplementary Fig. 2C). In mice treated with a mid dose of IiOVA LV, we observed a lower fraction of OVA-specific TEX CD8+ T cells (Supplementary Fig. 2D). Additionally, we noted a positive trend in the fraction of OVA-specific PEX CD8+ T cells in the IiOVA mid dose compared to the high dose-treated animals. Moreover, livers from the IiOVA high dose group displayed a lower number of integrated LV copies per genome compared to Control, although injected with the same dose, indicating that upon CD8+ T cell activation, antigen-expressing APCs might be depleted, as observed in other studies[22] (Supplementary Fig. 2E). These findings underscore the importance of achieving an optimal antigen load to effectively activate OVA-specific CD8+ T cells in the liver and maintain a functional balance between expansion of TEX and PEX subsets. Building on this observation, all further experiments were performed employing the delivery of a mid-dose of TA LV.

We then investigated the effect of IiOVA presentation by liver macrophages on established MC38.OVA LM. To this aim, we first implanted the MC38.OVA tumors in the liver of C57BL/6 mice, and after 7 days, we left mice untreated (i.e., Control) or treated them with IiOVA LV (Fig. 1G). We expected IiOVA treatment to delay tumor growth, but we found that IiOVA LV had no effect compared to Control (Fig. 1H). In agreement with this observation, MC38 clones expressing OVA were not cleared in the LM (Fig. 1I). However, IiOVA treatment expanded OVA-specific CD8+ T cells in circulation and in the liver, mainly with a TEX phenotype, but did not favor their infiltration in the tumor (Fig. 1, J, K and Supplementary Fig. 2F−H).

This observation indicates that in the presence of established LM, expansion of TA-specific CD8+ T cells in the liver and circulation might not be sufficient to significantly delay tumor growth and clear TA-expressing cancer cell clones.

To improve TA-specific CD8+ fitness and reprogram the TME, we employed *Mrc1*-driven LVs to enforce the expression of immune-activating cytokines in liver macrophages and TAMs in coordination with TA presentation. To this aim, we employed *Mrc1*-driven LVs expressing IFNα (IFNα LV)[15], or IL-12p40/IL-12p35 fusion cytokine (IL-12 LV, Fig. 1L). We treated mice bearing established MC38.OVA LM with (1)

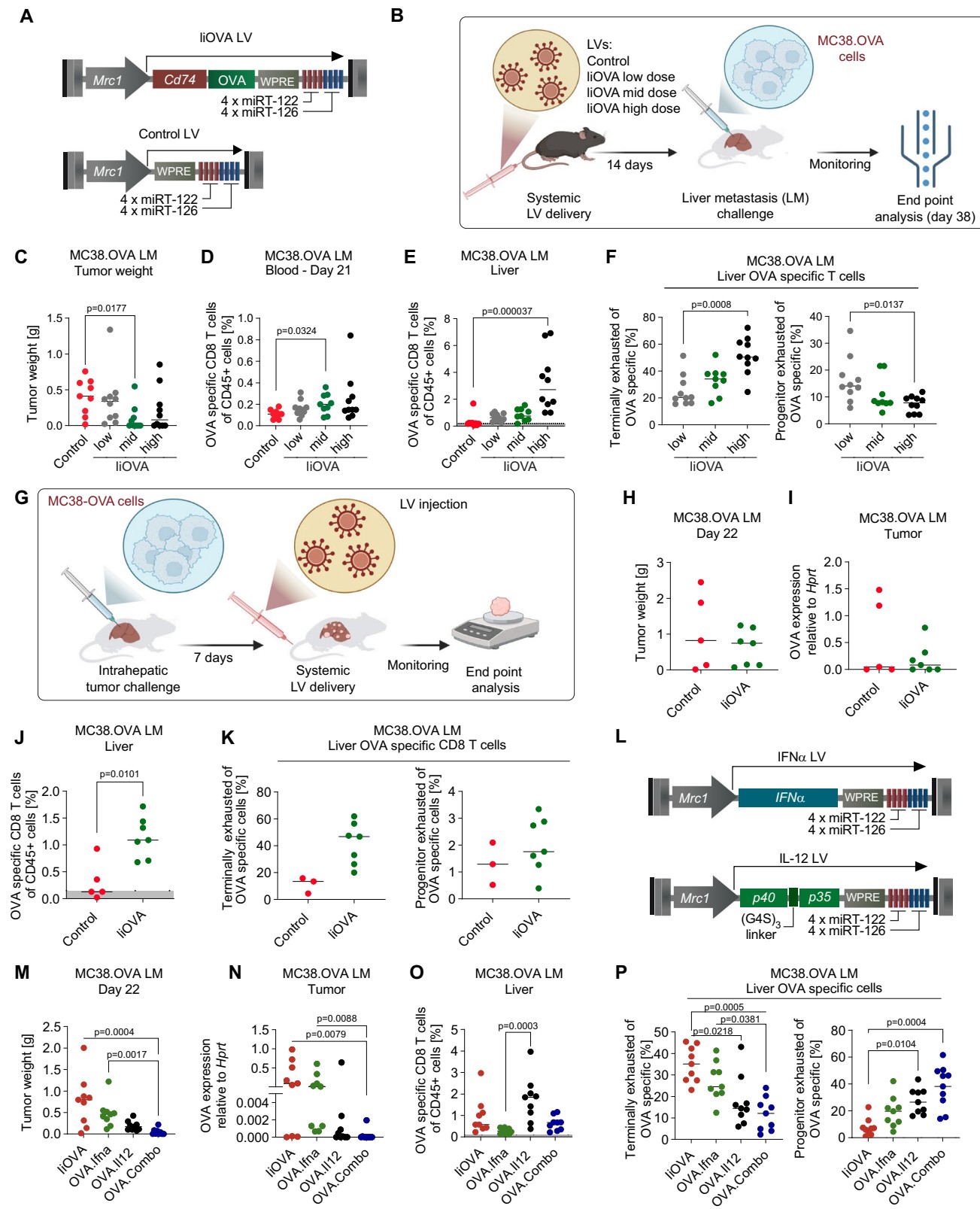

IiOVA LV; (2) IiOVA LV + IFNα LV (OVA.Ifna group); (3) IiOVA LV + IL-12 LV (OVA.Il12 group); or (4) IiOVA LV + IL-12 LV + IFNα LV (OVA.Combo group). When concurrently delivering LVs encoding cytokines and TAs, we mixed the respective LV preparations before intravenous injection. Cytokine co-delivery with IiOVA LV resulted in enhanced tumor control, with the OVA.Combo group displaying the highest therapeutic response (Fig. 1M and Supplementary Fig. 2I). Of note, 4

out of 9 mice eradicated LM in the OVA.Combo group. Moreover, in the remaining 5 out of 9 LM, we found a virtually complete clearance of OVA-expressing cancer cell clones, suggesting improved OVA reactive T cell function (Fig. 1N).

In agreement with the expression of IFNα and IL-12 from trans-duced liver macrophages, we detected IFNα (~200–300 pg/ml) and/or IL-12 (~100 pg/ml) in the plasma of the mice after one week from

**Fig. 1 | Coordinated IFNα and IL-12 expression rescues tumor-reactive T-cell activity. A** Schematic of the LVs employed. **B** Schematic of the experiment in (**C–F**). Illustration from BioRender. **C–F** Delivery of control LV or IiOVA LV before intra-hepatic tumor challenge (Control LV 10⁸ TU/mouse, low dose 10⁶ TU/mouse, mid-dose 10⁷ TU/mouse, high dose 10⁸ TU/mouse). **C** Liver metastasis weight ($n = 9, 10, 9, 10$ mice/group for Control, IiOVA low, IiOVA mid, IiOVA high, horizontal line represents median, statistical analysis by Kruskal–Wallis with Dunn's tests, $p \leq 0.05$ are shown). **D** Percentage of OVA-specific CD8⁺ T cells measured by flow cytometry (FC). Number of mice and statistics as in **C**. **E** FC analysis of the liver. Values below the dark gray area are excluded from further analyses (number of mice and statistics as in **C**). **F** FC analysis of the liver ($n = 10, 9, 10$ mice/group for IiOVA low, IiOVA mid, IiOVA high respectively, horizontal line represents median, statistical analysis by Kruskal–Wallis with Dunn's tests, $p \leq 0.05$ are shown). **G** Schematic of the experiments in **H–K** and **M–P**. Illustration from BioRender. **H, I** Therapeutic delivery of Control LV or mid dose IiOVA (10⁷ TU/mouse). **H** Tumor weight ($n = 5, 7$

mice/group for Control and IiOVA, respectively, horizontal line represents median, statistical analysis by Mann–Whitney test, $p \leq 0.05$ are shown). **I** OVA gene expression analysis performed on residual LM (number of mice and statistics as in **H**). **J, K** FC analysis of the liver. For OVA-specific CD8⁺ T cells, values below the dark gray area are excluded from further analyses ($n = 3, 7$ mice/group). **L** Schematics of the LVs employed to deliver cytokines. **M–P** Treatment with IiOVA, OVA.Ifna OVA.Il12 or OVA.Combo LV after 7 days post MC38.OVA tumor challenge (IiOVA $1 \times 10^7$ TU/mouse, OVA.Ifna total dose $1.1 \times 10^8$ TU/mouse, OVA.Il12 total dose $1.1 \times 10^7$ TU/mouse OVA.Combo total dose $1.2 \times 10^8$ TU/mouse). **M** Tumor weight ($n = 9$ mice/group, horizontal line represents median, statistical analysis by Kruskal–Wallis with Dunn's tests, $p \leq 0.05$ are shown). **N** OVA gene expression analysis performed on residual LM (number of mice and statistics as in **M**). **O, P** FC analysis of the liver. For OVA-specific CD8⁺ T cells, values below the dark gray area are excluded from further analyses (number of mice and statistics as in **M**).

---

OVA.Ifna, OVA.Il12 or OVA.Combo treatment, respectively (Supplementary Fig. 2J).

In the blood, OVA.Combo increased the fraction of circulating OVA-specific CD8⁺ T cell clones (Supplementary Fig. 2K). Interestingly, in all groups, most OVA-specific CD8⁺ T cells expressed PD1, a marker associated with activation and/or exhaustion. However, OVA.Combo treatment reduced the MFI of PD1 on these cells. (Supplementary Fig. 2L). OVA.Ifna and OVA.Combo increased the expression of the activation marker Ly6c compared to OVA.Il12 or IiOVA alone. The reduction of PD1 and increase in Ly6c expression might suggest an increased activation of OVA-specific CD8⁺ T cells in the circulation of OVA.Combo-treated mice.

In the liver, OVA-specific CD8⁺ T cells were detected in all groups, reaching a higher percentage in OVA.Il12 group (Fig. 1O). Importantly, addition of any cytokine favored the development of OVA-specific PEX CD8⁺ T cells while reducing the generation of OVA-specific TEX CD8⁺ T cells, with OVA.Combo, leading to the highest effect (Fig. 1P). Besides influencing OVA-specific T-cell phenotype, cytokines also affected the phenotype of other bystander CD8⁺ T cells (CD8⁺ H2Kb/SIINFEKL tetramer^neg). OVA.Il12 and OVA.Combo increased the fraction of bystander and TEX bystander CD8⁺ T cells compared to the IiOVA group, with OVA.Il12, leading to the strongest effect (Supplementary Fig. 2M, N). However, the effect of the cytokines on the phenotype of bystander CD8⁺ T cells was inferior to the one observed on OVA-specific CD8⁺ T cells. The treatment also affected the phenotype of CD4⁺ T cells. OVA.Il12 and OVA.Combo reduced the fraction of CD4⁺ T cells in the liver, whereas they increased the expression of the activation/exhaustion marker PD1 (Supplementary Fig. 2O).

In the residual LM, OVA.Il12 and OVA.Combo enhanced the infiltration of CD8⁺ T cells, particularly of OVA-specific CD8⁺ T cells, compared to IiOVA treatment (Supplementary Fig. 2P). OVA.Il12 increased the fraction of OVA-specific PEX CD8⁺ T cells and reduced the fraction of TEX CD8⁺ T cells (Supplementary Fig. 2Q). Notably, in the only two mice with analyzable tumors in the OVA.Combo group, we observed the highest fraction of OVA-specific PEX CD8⁺ T cells and lowest fraction of OVA-specific TEX CD8⁺ T cells. Contrary to the observation in the healthy liver parenchyma, in residual LM OVA.Il12 and OVA.Combo increased the fraction of CD4⁺ T cells (Supplementary Fig. 2R). Of note, both OVA.Il12 and OVA.Combo groups displayed the lowest percentage of Foxp3⁺ CD25⁺ CD4⁺ Treg cells (Supplementary Fig. 2S). Altogether, these results support the positive effect on T cell reprogramming of liver macrophages presenting TAs in combination with the release of immune-activating cytokines. Similar results, including the therapeutic effects, were observed when treating mice with the LVs earlier after tumor injection (Supplementary Fig. 3A–D). Treatment was well tolerated by mice as evidenced by stable mouse weight and normal transaminase levels (ALT/AST) throughout the experiment (Supplementary Fig. 3D, E). To investigate whether enforced expression of IFNα and IL-12 was associated with enhanced release of other inflammatory cytokines, we performed a multiplex

cytokine analysis. Interestingly, we found increased plasma levels of chemokines CCL3, CCL4, and CCL11 in mice treated with OVA.Combo (Supplementary Fig. 3F, G). CCL3, CCL4, and CCL11 have been associated with the recruitment of tumors of immune cells, including macrophages and T cells[23,24]. Notably, other pro-inflammatory cytokines typically associated with IL-12 induced systemic toxicity, such as TNFα, IFNγ or IL-6, were not significantly elevated by any of the treatments compared to IiOVA[25,26].

We then investigate to what extent the expression of cytokines and TA in the spleen were involved in driving anti-tumoral immune responses. To this aim, we surgically remove the spleen of the mice before tumor implantation and LV treatment. OVA.Combo treatment impaired tumor growth up to complete tumor regression in some mice and enabled the expansion of OVA-specific CD8 + T cells in circulation and in the liver of mice compared to controls (Supplementary Fig. 3H–J). Expanded OVA-specific CD8 + T cells displayed a phenotype consistent with earlier experimental results in unsplenectomized mice. These data suggest that the therapeutic effect of the simultaneous delivery of TA and cytokines is independent of the residual expression of the transgenes in splenic macrophages.

In summary, the coordinated expression of TAs with IFNα, and IL-12 outperformed TA expression alone or in combination with a single cytokine, resulting in complete tumor eradication in most mice without inducing severe toxicity. This effect was linked to an increased activation of OVA-specific CD8⁺ T cells, which exhibited a PEX phenotype.

## Concurrent IFNα and IL-12 expression increases MHC-I and MHC-II presentation in LMs

To better investigate the effect of simultaneous IFNα and IL-12 delivery on immune cells, we performed single-cell RNA sequencing (scRNA-seq) on CD45⁺ cells isolated from healthy liver parenchyma (Fig. 2A) and matched LMs of treated mice (Fig. 2B, Supplementary Fig. 4A). We identified and annotated distinct cell types based on their transcriptomic profile (Supplementary Fig. 4B–E, and Supplementary data 1). Within the APC compartment, we identified KCs, macrophages, monocytes, granulocytes (neutrophils and basophils), and different subsets of DCs (Fig. 2C, D). Analyzing together all APC cell types, we observed in both the liver and LM that OVA.Ifna increased the expression of genes involved in MHC-I antigen presentation, such as MHC-I subunits (*H2-Q7, H2-T22, H2-Q4, H2-K1, H2-M3, H2-D1*) and peptide transporters (*Tap1, Tap2*), and of interferon-stimulated genes (*Ifi44, Irf7, Isg20, Oas1a, Oas1g*) compared to the IiOVA group (Fig. 2E, F). On the other hand, OVA.Il12 enhanced the expression of genes associated with MHC-II antigen presentation, such as the MHC-II subunits (*H2-Eb1, H2-Ab1, H2-Aa, H2-DMb1, H2-Dma*) and the MHC-II transactivator *Ciita*, as well as genes associated to IFNγ stimulation (*Ccl5, Upp1, Slamf7, Gbp4, Cd74*), compared to the IiOVA group. In agreement with a superior therapeutic effect, OVA.Combo treatment increased the expression of genes involved in both MHC-I and MHC-II

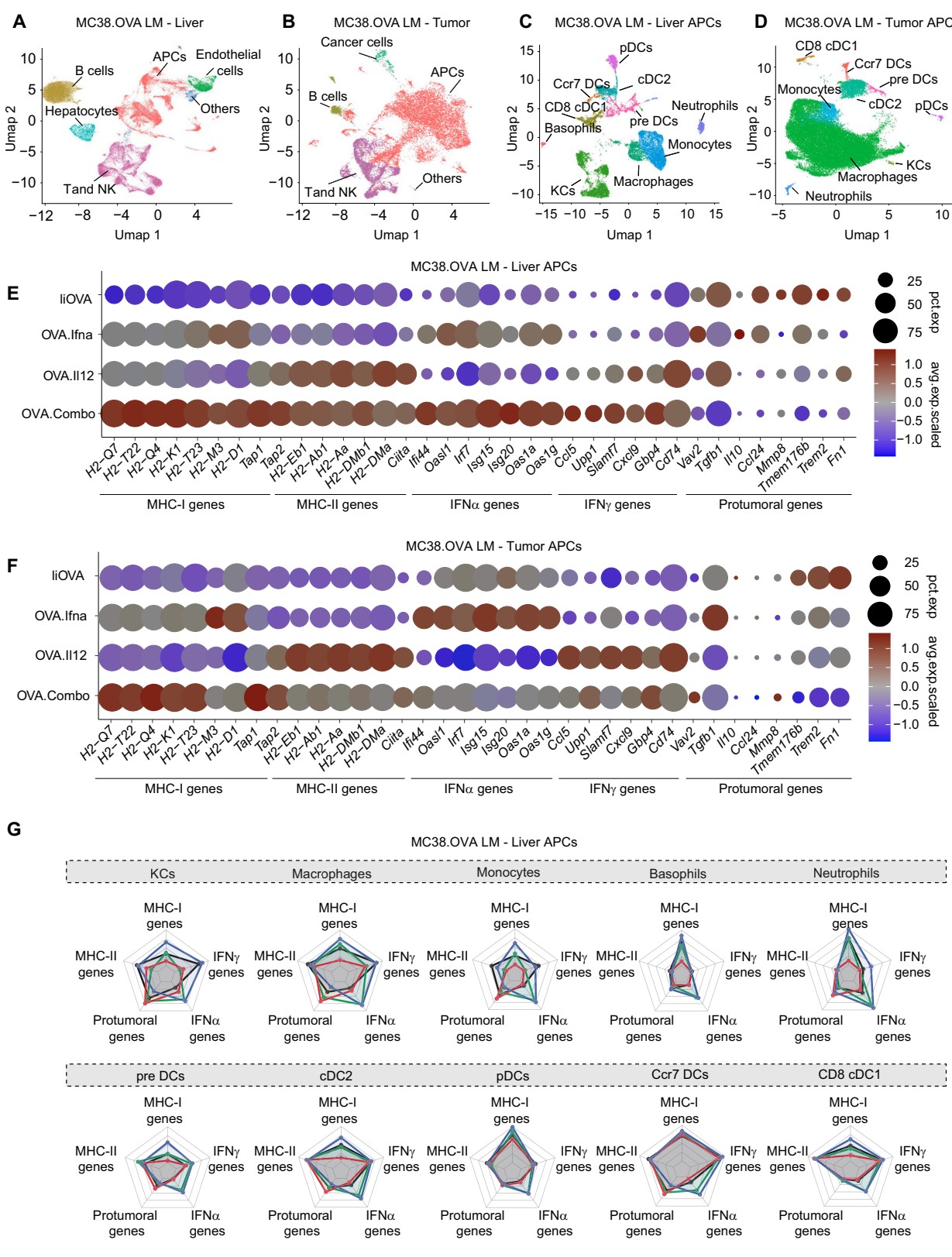

**Fig. 2 | Concurrent IFNα and IL-12 expression increases MHC-I and MHC-II presentation in liver metastases. A**, **B** UMAP projection of single-cell RNA sequencing (scRNA-seq) data obtained from CD45+ cells sorted from a healthy liver lobule (**A**) or a matched liver metastasis (**B**) from mice treated with liOVA, OVA.Ifna, OVA.Il12 or OVA.Combo (*n* = 2, 3, 3, 3 mice/group for liOVA, OVA.Ifna, OVA.Il12, and OVA.Combo, respectively). **C**, **D** UMAP projection of scRNA-seq of APC subcluster for the indicated tissues. **E**, **F** Fraction of positive cells and scaled averaged expression of selected genes belonging to the indicated biological processes in the APCs from indicated tissues (*n* = 2, 3, 3, 3 mice/group for liOVA, OVA.Ifna, OVA.Il12, and OVA.Combo). **G** Combined gene expression score for genes belonging to the indicated categories in the distinct liver myeloid populations.

antigen presentation, as well as IFNα and IFNγ signaling. Moreover, in APCs from OVA.Combo-treated animals we observed a decreased expression of genes associated with immunosuppressive and pro-tumoral functions of APCs (*Vav2, Tgfb1, Il10, Ccl24, Mmp8, Tmem176B, Trem2, Fn1*) in both liver and LM. We then investigated how distinct cell types among those identified in the APC compartment were affected by the treatment. On KCs, macrophages, and monocytes, OVA.Combo and OVA.Il12 preferentially upregulated genes related to MHC-II presentation and IFNγ signaling (Fig. 2G and Supplementary Fig. 4F). Conversely, in the same cell types, OVA.Combo and OVA.Il12 downregulated the expression of pro-tumoral and immunosuppressive genes. OVA.Combo and OVA.Ifna promoted the expression of genes associated with MHC-I presentation and IFNα signaling in distinct subsets of DCs. In agreement with these findings, gene-set enrichment analysis (GSEA) showed that categories related to immune activation, such as antigen processing and presentation, antigen binding, MHC-II presentation, and positive regulation of cell killing were enriched in KCs, macrophages and monocytes in the OVA.Combo group compared to IiOVA in the liver and LM (Supplementary Fig. 5A, B).

OVA.Combo or the single cytokines and TA alone did not impact the general composition of TAMs subsets in the TME as compared to TA only (Supplementary Fig. 5C, D). This observation may be associated with the presence of IiOVA in the control group, which, per se, could drive the expansion of inflammatory TAM clusters indirectly through CD8⁺ T cell activation. However, LV-based expression of either IL-12 or IFNα, and to a higher extent their combination, significantly modified the phenotype of TAMs, by increasing the expression of genes associated with antigen presentation (both MHC-I and MHC-II) and cytokine stimulation, while reducing the expression of pro-tumoral genes in most of the subsets we defined (Supplementary Fig. 5E). Furthermore, we validated these findings on tumor myeloid cells, including TAMs, by employing multicolor flow cytometry (FC, Supplementary Fig. 1C). This analysis confirmed enhanced expression of MHC-II molecules, co-stimulatory receptors as well as other markers of immune activation (Supplementary Fig. 5F–K).

In summary, simultaneous delivery of IiOVA, IFNα and IL-12 promoted reprogramming of liver and tumor APCs, including KCs, macrophages and monocytes, boosting their antigen presenting functions and reducing the expression of pro-tumoral genes.

## Concurrent IFNα and IL-12 expression ameliorates the fitness of CD4⁺ and TA-specific CD8⁺ T cells

Within the T and NK cluster, we identified and manually annotated distinct populations of CD4⁺, CD8⁺, NK, and innate lymphoid cells in liver and matched LM based on their expression profile (Fig. 3A and Supplementary Fig. 6A, and Supplementary data 2). By performing GSEA, we found that IFNα increased the expression of genes associated with negative regulation of viral replication as well as IFNα and IFNγ signaling on T and NK cells compared to the IiOVA group (Fig. 3B). On the other hand, IL-12 increased the expression of genes associated with enhanced cell division and proliferation. The combination of IL-12 and IFNα in the OVA.Combo group resulted in an additive effect leading to the upregulation of all these gene ontology (GO) terms. Interestingly, focusing on liver CD8⁺ T cells, OVA.Ifna increased the expression of genes associated with IFNα signaling, such as *Stat1, Ly6c2, Oas1a, Irf8, Ifitm3*, and PEX phenotypes such as *Tbx21, Klrg1, Cx3cr1, Tcf7, Klf2*, and *Sell*, while it reduced the expression of genes associated with T cell exhaustion, such as immune checkpoint molecules (*Pdcd1, Lag3, Ctla4, Havcr2, Tigit*), IL-10 receptor *Il10ra* and the transcriptional regulator of exhaustion, *Tox*. On the other hand, OVA.Il12 increased the expression of genes associated with IFNγ stimulation, such as cytokine receptors (*Il18rap, Il18r1, Il12rb1*) and IFNγ response genes (*Ifngas1, Gbp2*), as well as genes associated with effector functions such as granzymes (*Gzma, Gzmb*), perforin (*Prf1*), inflammatory cytokines (*Ifng, Tnf*) and death-inducing signaling *FasL*. Genes associated with T cell exhaustion were

upregulated by OVA.Il12 treatment but the combination of IL-12 with IFNα in the OVA.Combo group reduced exhaustion compared to OVA.Il12 and increased the expression of genes associated with effector functions and PEX T cells compared to IiOVA alone, suggesting additive effects of these two cytokines (Fig. 3C, D).

To identify TA-specific CD8⁺ T cells among CD45⁺ cells in the scRNA-seq dataset, we stained OVA-specific CD8⁺ T cells with DNA-barcoded antibodies. DNA-barcoded CD8⁺ T cells were isolated independently and pooled with all CD45⁺ cells. We also employed TCR sequencing to track distinct T cell clonotypes, including OVA-specific and bystander CD8⁺ T cell clones, across liver and matched LM. Most OVA-specific clones were shared across the liver and the LM in all groups (Supplementary Fig. 6B) and enriched in the cluster of CD8⁺ Teff3, indicating that virtually all OVA-specific CD8⁺ T clones were activated (Fig. 3A, E). However, genes associated with IFNα, IFNγ, and immune activation were upregulated in OVA-specific CD8⁺ T cells in the OVA.Combo group compared to other groups (Fig. 3F). In addition to this, we observed upregulation of PEX genes and downregulation of the exhaustion signature in the OVA.Combo groups compared to the IiOVA group (Fig. 3G and Supplementary Fig. 6C). OVA.combo effects appeared additive when compared to individual cytokine delivery. These findings suggest that IiOVA treatment alone can promote OVA-specific CD8⁺ T cell expansion in the liver, but only cytokine co-delivery reprograms their phenotype and favors their effector function.

Interestingly, clonotype tracking of CD4⁺ T cells revealed TCR sharing between clonally expanded CD4⁺ T cells in the liver and LM only in the mice treated with either OVA.Il12 or OVA.Combo (Fig. 3H). CD4⁺ T cell clonotypes shared between liver parenchyma and metastasis were enriched in the IFNγ CD4 cluster indicating a Th1 skewed state (Fig. 3A, I), which has been previously associated with response to immunotherapy and immune activation[27]. These cells upregulated genes associated with IFNγ stimulation (*Il18rap, Il18r1, Il12rb1* and *Ifng*) and effector functions (*Tnf, Il21, Gzmk, Fasl* and *Slamf1*), while downregulated genes associated with T cell exhaustion and immune suppression (Supplementary Fig. 6D). In agreement with this observation, OVA.Il12 and OVA.Combo reduced the fraction of Th2-skewed *Il4* CD4⁺ T cells and increased the number of small, large, and hyperexpanded CD4⁺ T cell clones compared to the other groups (Fig. 3J, K).

In summary, OVA.Combo therapeutic activity was associated with increased PEX phenotype, reduced exhaustion of OVA-specific CD8⁺ T cells, and increased clonal expansion and Th1 skewing of CD4⁺ T cells. The higher activation and expansion of CD4⁺ T cells observed in the liver agree with the enhanced MHC-II restricted antigen presentation observed in the OVA.Il12 and OVA.Combo groups compared to the other groups.

## Coordinated delivery of IFNα and IL-12 activates TA-specific CD8⁺ T cells independently of CD4⁺ T-cell help

We next investigated the contribution of CD4⁺ T cells to TA reactive CD8⁺ T cell activation upon treatment. We first assessed the impact of the MHC-II-restricted moiety in the TA LV design for activating and expanding TA-specific CD8⁺ T-cell in the liver. We generated an *Mrc1*-driven LV enabling the expression of a truncated form of chicken ovalbumin (dOVA), which, according to previous reports, results in the virtually exclusive presentation of OVA-derived peptides in MHC-I, but not in MHC-II complexes, thus not engaging CD4⁺ T cells[16]. Furthermore, to assess whether CD74 fusion could impact the observed effects, we generated a variant of OVA consisting of amino acids 242–277 fused to CD74 as we did for IiOVA (Supplementary Fig. 7A). This variant encompasses the OVA-associated immunodominant MHC-I restricted peptide SIINFEKL but lacks both known or predicted immunodominant MHC-II-restricted peptides. We also employed an LV driving the expression of CD74 fused with SIINFEKL and with an immunodominant MHC-II-restricted peptide found in mycobacterium tuberculosis (TB) hereon indicated as SIINFEKL.TB. Of note, TB is not

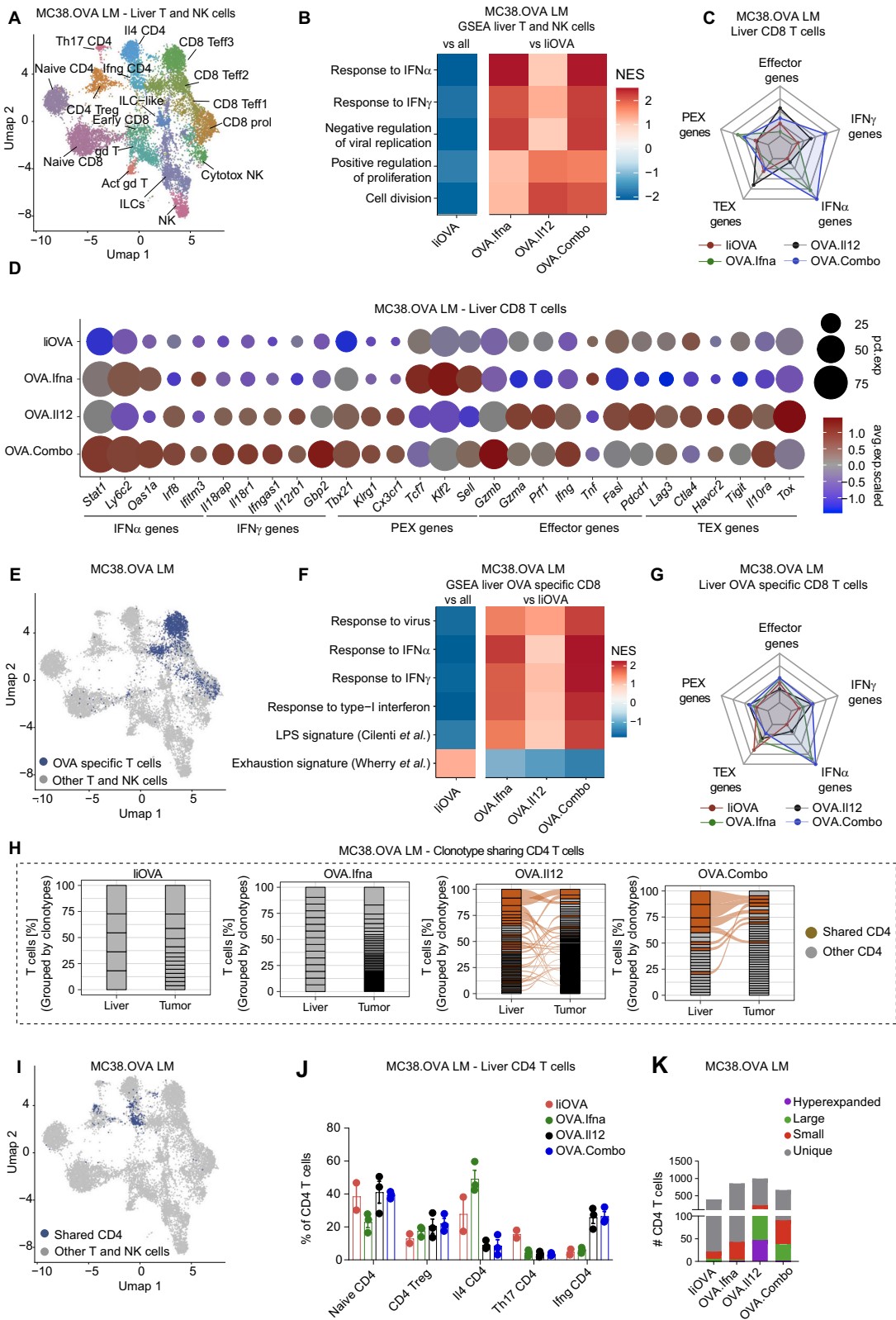

expressed by MC38.OVA cancer cells, however according to previous studies, presentation of the immunodominant TB peptide by APCs enables CD4[+] T cell helper function[28,29]. We assessed the ability of the distinct TA LVs to induce protective immunity against a peripheral tumor challenge. Fourteen days after treatment with low doses of dOVA, SIINFEKL, or SIINFEKL.TB LV, we implanted s.c. MC38.OVA cells

(Supplementary Fig. 7B, C). We found that the addition of the MHC-II-restricted peptide TB increased immune activation in the SIINFEKL.TB cohort, leading to delayed tumor growth and higher clearance of OVA cancer cell clones in the s.c. tumors compared to the dOVA and SIINFEKL LV treatments (Fig. 4A, B). In all the treated mice we observed expansion of OVA-specific CD8[+] T cells in the liver, but notably only

**Fig. 3 | Concurrent IFNα and IL-12 expression ameliorates the fitness of CD4+ and TA-specific CD8+ T cells. A** UMAP projection of scRNA-seq of liver T and NK cells subclustered. **B** Gene set enrichment analysis (GSEA) of scRNA-seq data showing normalized enriched score (NES) for selected gene ontology (GO) terms calculated based on genes differentially expressed in T and NK cells in the indicated comparisons ($n = 2, 3, 3, 3$ mice/group for IiOVA, OVA.Ifna, OVA.Il12 and OVA.-Combo). **C** Combined gene expression score for genes belonging to the indicated categories in liver CD8+ T cells. **D** Fraction of positive cells and scaled averaged expression of selected genes belonging to the indicated biological processes in the liver CD8+ T cells (number of mice as in (**B**)). **E** UMAP projection of liver scRNA-seq indicating cells bearing OVA-specific TCRs. **F** GSEA of scRNA-seq data showing NES

for selected GO terms calculated based on genes differentially expressed in OVA-specific CD8+ T cells in the indicated comparisons (number of mice as in **B**). **G** Combined gene expression score of genes belonging to the indicated categories in OVA-specific CD8+ T cells. **H** Clonotype sharing between liver and tumor CD4+ T cells, grouped by TCR clonotype. **I** UMAP projection of liver scRNA sequencing indicating CD4+ T cells clonotypes shared between liver and tumor tissue. **J** Percentage of cells within the CD4+ T cells populations in the liver ($n = 2, 3, 3, 3$ mice/group for IiOVA, OVA.Ifna, OVA.Il12, and OVA.Combo, horizontal line represents mean, error bars represent the SEM). **K** Number of CD4+ T cells divided by TCR clonotype frequency.

treatment with SIINFEKL.TB reduced the number of TEX CD8+ T cells, whereas it produced a positive trend in the number of PEX CD8+ T cells (Fig. 4C, D). This was associated with a significant increase in the infiltration of OVA-specific CD8+ T cells in the tumor of SIINFEKL.TB-treated mice (Supplementary Fig. 7D). These results confirm the importance of MHC-II presentation in liver resident macrophages to induce TA-specific CD8+ T cell activation with protective functions.

To understand whether IFNα and IL-12 co-delivery could overcome the need of engaging CD4+ T cells to efficiently activate TA specific CD8+ T cells, we treated mice bearing MC38.OVA LM with IFNα and IL-12 LVs in combination with either IiOVA LV (OVA.Combo) or SIINFEKL LV (SIIN-FEKL.Combo). We found that combination of IFNα and IL-12 reduced LM growth compared to untreated controls, leading to complete clearance of OVA-expressing MC38 clones in several mice independently of the presence of the MHC-II moiety in the delivered TA (Fig. 4E, F). In agreement with expression of IFNα and IL-12 from liver macrophages, we detected IFNα and/or IL-12 in the plasma of the mice after one week from treatment (Supplementary Fig. 7E). Similarly to OVA.Combo, SIINFEKL.-Combo treatment increased the fraction of circulating OVA-specific CD8+ T cells compared to control mice (Supplementary Fig. 7F, G).

In the liver, we found a comparable percentage of OVA-specific CD8+ T cells between groups (Fig. 4G). Interestingly, we measured a similar percentage of PEX and TEX OVA-specific CD8+ T cells in OVA.-Combo- and SIINFEKL.Combo-treated mice (Fig. 4H). To further capture differences between OVA-specific CD8+ T cells induced upon treatment with IiOVA, SIINFEKL, OVA.Combo or SIINFEKL.Combo, we performed scRNA-seq analysis of liver CD45+ cells enriched with DNA-barcoded OVA-specific CD8+ T cells. Of note, analyses were performed only on liver cells since LM were virtually absent in most of the treated mice. In OVA-specific CD8+ T cells from OVA.Combo and SIINFEKL.-Combo-treated mice, scRNA-seq showed (1) upregulation of genes involved in IFNα and IFNγ signaling; (2) increased expression of PEX T cell-associated genes; and (3) downregulation of TEX T cell-associated genes (Fig. 4I and Supplementary Fig. 7H). In agreement with their activation and with our previous findings, all OVA-specific CD8+ T cells were associated with the CD8+ Teff3 cluster, composed by effector/exhausted-like CD8+ T cells (Supplementary Fig. 7I, J). Interestingly, CD4+ T cells in the OVA.Combo and SIINFEKL.Combo group upregulated the expression of genes associated with IFNα/IFNγ signaling and Th1-skewing. On the other hand, genes associated with Th2/Th17-skewing were downregulated in these groups compared to IiOVA-treated animals (Fig. 4J, K). Moreover, either OVA.Combo or SIIN-FEKL.Combo treatment promoted expansion of CD4+ T cell clonotypes, independently of the presence of the MHC-II moiety in the TA expressed from the liver-directed LV (Fig. 4L). Expanded CD4+ T cells were mostly associated with the IFNγ CD4+ T cell cluster and displayed an activated Th1-skewed phenotype (Supplementary Fig. 7I, K). In agreement with our previous findings, APCs from Combo-treated groups upregulated the expression of genes involved in both MHC-I and MHC-II antigen presentation, as well as IFNα and IFNγ signaling (Supplementary Fig. 7L, M). Overall, these data support the idea that IFNα and IL-12 activate OVA-specific CD8+ T cells and CD4+ T cells independently of the presence of MHC-II-restricted peptides in the delivered TA.

CD4+ T cells have been previously found to be necessary to enable effector CD8+ T cell functions in chronic infection and cancer[30,31]. To assess whether CD4+ T cells as well as other immune cells are essential for the therapeutic effects of OVA.Combo, we conducted selective depletion of CD4+ T cells, CD8+ T cells, and NK cells using monoclonal antibodies (Supplementary Fig. 8A). Interestingly, depletion of CD8+ T cells increased plasma levels of IFNα and IL-12, suggesting that, as previously observed, CD8+ T cells kill OVA-expressing APCs (Supplementary Fig. 8B). Notably, depletion of CD8+ T cells completely abrogated the therapeutic effects of OVA.Combo treatment, while CD4+ T cell or NK cell depletion did not (Fig. 4M, N). In agreement with this finding, we observed that depletion of CD4+ T cells or NK cells did not modify the fraction or phenotype of circulating OVA-specific CD8+ T cells (Supplementary Fig. 8C). Interestingly, liver OVA-specific CD8+ T cells matured and acquired features of PEX cells independently of the presence of CD4+ T cells (Fig. 4O, P).

Overall, these findings indicate that coordinated delivery of IFNα and IL-12 with TA significantly enhances TA-specific CD8+ T cell activation and tumor clearance without the need for MHC-II-restricted peptides in the delivered TA. While both CD4+ and CD8+ T cells are activated by the treatment, only CD8+ T cells are required to achieve TA-expressing cancer cell clearance, confirming that, when TA and cytokines are co-delivered, robust anti-tumor immunity is achieved independently of CD4+ T cell help.

## Concurrent delivery of naturally occurring TAs together with IFNα and IL-12 inhibits melanoma and CRC LM growth by expanding and reprogramming TA reactive CD8+ T cells

We then investigated the effect of IFNα and IL-12 co-delivery with a naturally occurring TA. To do this, we employed a melanoma cell line, B16-F10, which expresses tyrosine-related protein 2 (TRP-2), a tumor-associated antigen (TAA) highly expressed in melanoma, with minimal expression in healthy tissues[32]. We designed a liver macrophage-targeting LV expressing the luminal region of TRP-2 fused to CD74 as we did to express IiOVA (Supplementary Fig. 9A). We then implanted B16-F10 melanoma cells in the liver of syngeneic mice to produce experimental LM. To understand if TA delivery is essential to foster TA-specific CD8+ T cells we treated LM-bearing mice with either cytokine LVs alone (i.e., IFNα + IL-12 LVs, hereon Ifna.Il12) or with TRP-2 LV in combination with cytokine LVs (hereon Trp2.Combo, Supplementary Fig. 9B). Plasma levels of IFNα and IL-12 post-treatment were similar to those observed in our previous experiments (Supplementary Fig. 9C). Magnetic resonance imaging (MRI) analysis showed therapeutic effect of both Ifna.Il12 and Trp2.Combo treatments compared to control untreated mice, with Trp2.Combo displays a stronger effect (Fig. 5A, B). Compared to the other groups, Trp2.Combo increased PD1 expression in circulating CD8+ T cells and the Ly6c+ CD44+ fraction of circulating CD8+ T cells (Supplementary Fig. 9D). Interestingly, these cells negatively correlated with the tumor volume, suggesting that these cells may comprise putative tumor-reactive CD8+ T cells (Supplementary Fig. 9E).

Compared to Control mice, we observed enhanced infiltration of CD8+ T cells, including TEX CD8+ T cells, in the livers of Trp2.Combo and Ifna.Il12-treated mice (Fig. 5C). However, in accordance with our findings

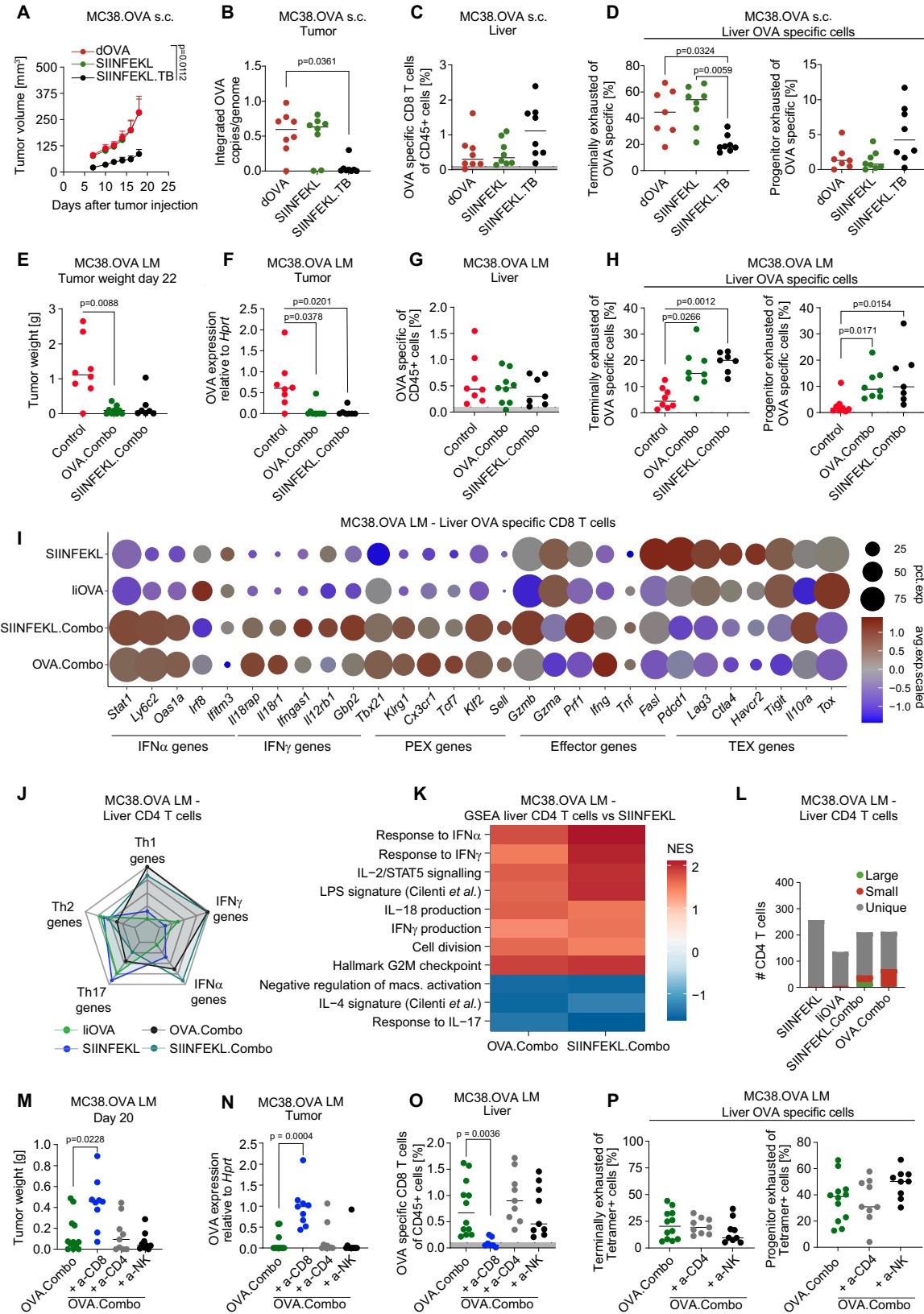

in the MC38.OVA model, we observed a higher fraction of CD8+ T cells with a PEX phenotype in Trp2.Combo mice. These observations indicate a superior activation and expansion of CD8+ T cells in the Trp2.Combo group. In a similar manner, in both the Ifna.Il12 and the Trp2.Combo groups, CD4+ T cells upregulated the expression of PD1, suggesting their activation (Fig. 5D). In agreement with this observation, LM from

Trp2.Combo and Ifna.Il12-treated mice displayed (1) enhanced infiltration of both CD4+ and CD8+ T cells; (2) increased infiltration of Th1 CD4+ T cells; and (3) reduced infiltration of Tregs compared to controls (Supplementary Fig. 9F–I).

To investigate whether the superior therapeutic efficacy observed in Trp2.Combo-treated mice compared to Ifna.Il12 was associated with

**Fig. 4 | Coordinated delivery of IFNα and IL-12 activates TA-specific CD8⁺ T cells independently of CD4⁺ T cell help. A–D** Delivery of dOVA, SIINFEKL or SIIN-FEKL.TB LV to mice before tumor implantation (10⁷ TU/mouse). **A** s.c. tumor growth (n = 8 mice/group, mean ± SEM; statistical analysis by two-sided Mann–Whitney test on the area under the curve, p ≤ 0.05 are shown). **B** LV copies per genome of the tumor by ddPCR analysis (number of mice as in **A**; horizontal line represents median; statistical analysis by Kruskal–Wallis with Dunn's tests, p ≤ 0.05 are shown). **C, D** FC analysis of the liver (number of mice and statistic as in **B**, values below the dark gray area are excluded from further analyses). **E–H** Treatment with, OVA.Combo or SIINFEKL.Combo 7 days after tumor challenge (total dose 1.2 × 10⁸ TU/mouse; Control mice are untreated). **E** Tumor weight (n = 8, 9, 7 mice/group for Control, OVA.Combo, or SIINFEKL.Combo respectively, horizontal line represents median, statistical analysis by Kruskal–Wallis with Dunn's tests, p ≤ 0.05 are shown). **F** OVA gene expression analysis on residual LM (number of mice and statistics as in **E**). **G, H** FC analysis of the liver. Values below the dark gray are excluded from

further analysis (number of mice and statistics as in **E**). **I** Expression of selected genes in liver OVA-specific CD8⁺ T cells (n = 1 mouse/group). **J** Combined gene expression score in liver CD4⁺ T cells (n = 1 mouse/group). **K** GSEA of scRNA sequencing data showing NES for selected GO terms in liver CD4⁺ T cells (n = 1 mouse/group). **L** Number of CD4⁺ T cells divided by TCR clonotype frequency (n = 1 mouse/group). **M–P** Mice were injected intrahepatically with MC38.OVA cells and, after 6 days, injected IP with a-CD4, a-CD8, or a-NK. On day 7 mice were injected with OVA.Combo (total dose 1.2 × 10⁸ TU/mouse) and then with the respective mabs IP twice weekly. **M** Tumor weight (n = 12, 9, 9, 9 mice/group for OVA.Combo, OVA.Combo + a-CD4, OVA.Combo + a-CD8, OVA.Combo + a-NK, horizontal line represents median, statistical analysis by Kruskal–Wallis with Dunn's tests comparing all groups vs OVA.Combo, p ≤ 0.05 are shown). **N** OVA gene expression analysis of residual LM (number of mice and statistics as in **M**). **O, P** FC analysis of the liver (number of mice and statistics as in **M**).

generation of tumor-specific CD8⁺ T cells, we performed an IFNγ ELISPOT assay on CD8⁺ T cells purified from the spleens of treated mice. CD8⁺ T cells were co-cultured with a thymoma cell line pulsed with the TRP-2 immunogenic peptide SVYDFFVWL. Notably, CD8⁺ T cells from 4 out of 5 Trp2.Combo-treated mice were activated by the TRP2 peptide compared to none in the Ifna.Il12 group (Fig. 5E). This result confirms that the superior tumor inhibition obtained by adding the TA to IFNα + IL-12 is achieved through the expansion of TA-specific CD8⁺ T-cell clones.

To investigate the efficacy of our platform in a translational setting in which TAs have not been previously identified, we employed AKTPF CRC cells derived from APC^D716; Kras^G12D; Tgfbr2^−/−; Trp53^R270H; Fbxw7^−/− mice[15,33]. To identify putative immunogenic peptides in this cell line, we performed whole exome sequencing (WES) and RNA sequencing of cultured AKTPF, AKTPF LM, and reference control healthy mouse tissue. By integrating two antigen prediction methods, antigen garnish and Pvac tools[34,35], we identified 33 putative peptides with predicted high binding affinity for the C57BL/6 H2-Kb MHC-I (Fig. 5F). Of note, building on the fact that our previous results suggest that MHC-II-restricted peptides are nor necessary in the presence of IFNα + IL-12 to drive effective immune activation, we focused our pipeline on the identification of MHC-I-restricted peptides. The 33 peptides identified in our analyses were combined into a single chimeric protein fused downstream to the CD74 moiety and incorporated in the liver macrophage-targeting LV, as we did to express IiOVA and TRP-2 (Fig. 5G). To reduce the possibility of generating immunogenic peptides from the junctions between different peptides, we arranged peptide order and incorporated linker sequences according to a previously described optimization method[34]. We then employed the resulting LV, hereon TA33, alone or in combination with LVs driving the expression of IFNα and IL-12, hereon TA33.Combo, to treat mice that were challenged with AKTPF CRC cells via intrasplenic injection (Fig. 5H). This approach enables spontaneous cancer cell seeding in the liver to generate LM. After LV delivery we detected levels of cytokines in the plasma of TA33.Combo-treated mice in line with our previous experiments (Supplementary Fig. 9J). TA33.Combo treatment impaired LM growth compared to untreated control or TA33-treated mice, leading to 6 out of 9 mice completely eradicating LM (Fig. 5I). TA33.Combo increased PD1 expression in circulating CD8⁺ T cells compared to control and led to a trend towards an increased fraction of effector Ly6c⁺ CD44⁺ CD8⁺ T cells, suggesting superior activation of CD8⁺ T cells compared to TA33 treatment (Supplementary Fig. 9K, L).

Compared to controls, in the livers of the TA33.Combo-treated mice we found higher infiltration of CD8⁺ T cells, which accounted for more than 20% of all CD45⁺ cells in the liver. Of note, both TA33 and TA33.Combo increased the fraction of TEX CD8 + T cells compared to controls. Conversely, as in the MC38.OVA and in the B16-F10 melanoma LM models, only TA33.Combo increased the fraction of CD8⁺

T cells displaying features of PEX T cells (Fig. 5J). Moreover, in accordance with our previous findings, we observed that TA33.Combo increased the expression of PD1 in CD4⁺ T cells compared to TA33 or control mice, suggesting increased activation of CD4⁺ T cells (Fig. 5K).

To investigate if TA33.Combo enabled the generation of tumor-specific CD8⁺ T cells, we performed an IFNγ ELISPOT assay on CD8⁺ T cells purified from the spleen of either control untreated or TA33.Combo-treated mice. Notably, CD8⁺ T cells isolated from all the TA33.Combo-treated mice responded to several pools of the predicted immunogenic peptides (Fig. 5L). Conversely, in the control group only one of the analyzed mice showed a limited response against one of the predicted immunogenic peptides (Fig. 5M).

To mirror the situation of patients who have had their primary tumors removed but still have circulating cancer cells, we treated the mice with TA33.Combo and then introduced cancer cells into the splenic circulation to enable spontaneous liver metastatic seeding. TA33.Combo protected the mice from metastatic dissemination in the liver, resulting in all treated mice being tumor-free for the duration of the experiment (Supplementary Fig. 10A–C). These mice displayed normal weight gain over the observed period and normal levels of transaminases compared to tumor-free control mice (Supplementary Fig. 10D, E). Hemocytometric analysis revealed a mild decrease in red blood cell, white blood cell, and lymphocyte count in mice treated with TA33.Combo compared to control, which was associated with an increase in the reticulocytes counts (Supplementary Fig. 10F, G). These parameters returned to levels comparable to tumor-free control mice toward the end of the experiment concomitantly with a decrease in cytokine levels observed in the plasma and clearance of transduced cells from the liver (Supplementary Fig. 10H, I). Histopathological evaluation of the liver confirmed the absence of LM and highlighted a slight increase in mixed cell inflammatory infiltrate and extramedullary hematopoiesis compared to tumor-free control mice (Supplementary Fig. 10J). These minimal alterations were also present in control mice, and might be attributed to mouse aging and animal housing conditions. Of note, we observed minimal to moderate inflammation in the splenic capsule and minimal to moderate lymphoid hyperplasia in a few mice in the LV-treated group. These findings may be at least in part associated with the surgical procedure that was employed to challenge the mice with experimental LMs through intrasplenic injection. Importantly, no significant alterations were observed in the lung, bone marrow, and mesenteric lymph nodes, confirming the safety and tolerability of this combination treatment.

Overall, these data support the superior therapeutic efficacy of coordinated TA, IFNα, and IL-12 delivery compared to IFNα and IL-12 alone and prove that, at the given dose, this therapeutic intervention is safe and well tolerated in mice and can induce the proliferation and expansion of TA reactive CD8⁺ T cells against naturally occurring TAAs or neoantigens, resulting in effective anti-tumor immunity against distinct mouse models of LM.

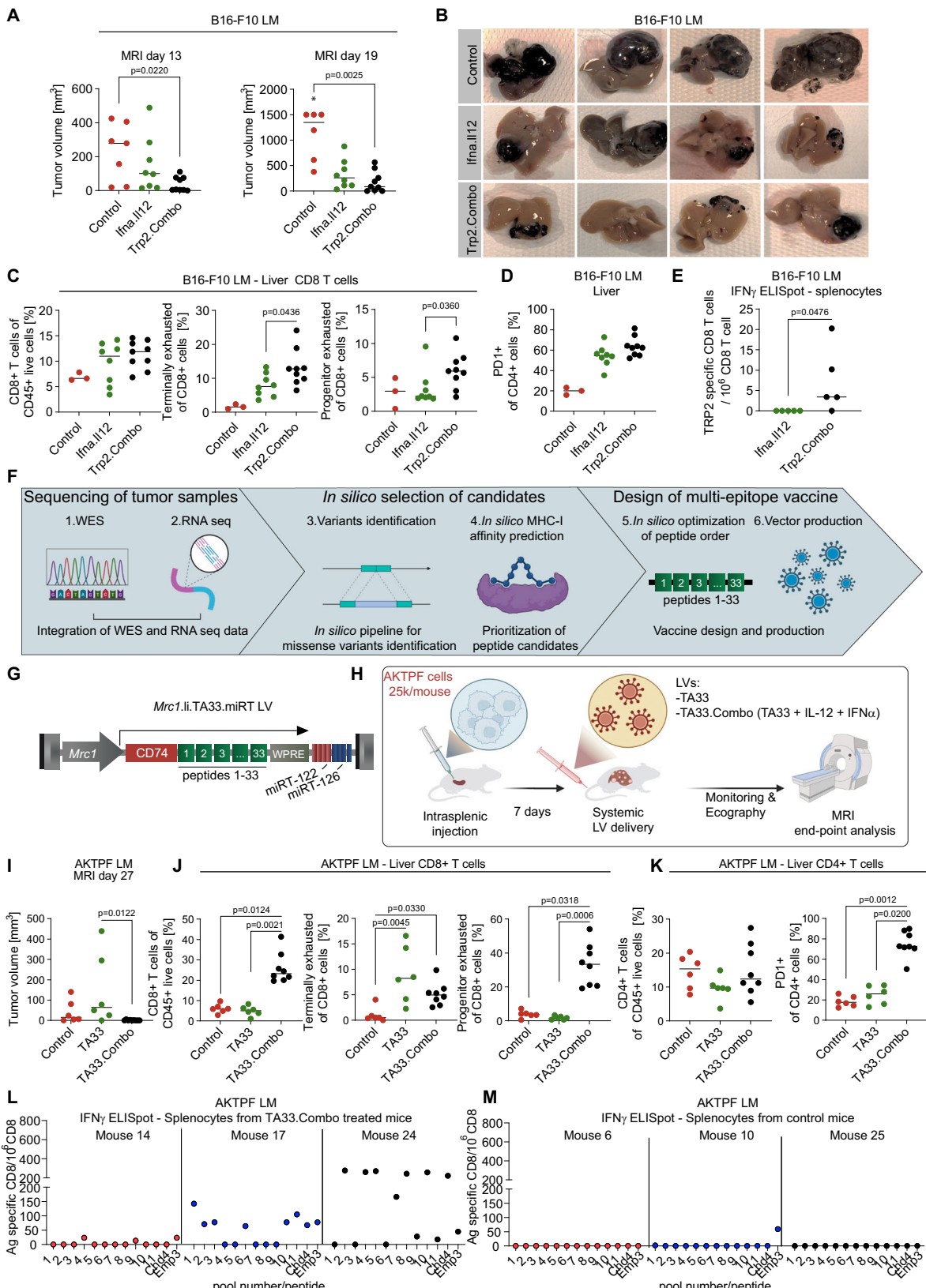

## IFNα and IL-12 rewire immune circuits within the metastatic microenvironment enabling TA-specific T-cell activity

To investigate the effects of IFNα and IL-12 on immune cells in AKTPF LMs, we performed scRNA and TCR sequencing on the matched liver and LM (Supplementary data 3 and supplementary data 4). APCs including monocytes, macrophages, granulocytes and DCs, upregulated genes involved in MHC-I and MHC-II-restricted antigen presentation or associated with IFNα and IFNγ signaling, and downregulated pro-tumoral genes, in both liver and LM upon TA33.Combo as compared to TA33 alone (Supplementary Fig. 11A–C). TA33.Combo promoted upregulation of genes related to MHC-II-restricted antigen presentation, especially in KCs, macrophages, and monocytes (Supplementary Fig. 11D).

**Fig. 5 | Concurrent delivery of naturally occurring TAs together with IFNα and IL-12 inhibits melanoma and CRC LM growth by expanding and reprogramming TA reactive CD8⁺ T cells. A–E** Treatment of mice bearing established B16-F10 LM with Ifna.Il12 or Trp2.Combo 5 days post tumor challenge (total dose of LVs: Ifna.Il12 1.1 × 10⁸ TU/mouse, Trp2.Combo 1.2 × 10⁸ TU/mouse). **A** Quantification of LM volume by magnetic resonance imaging (MRI) at the indicated time points (n = 7, 8, 9 mice/group, for Control untreated, Ifna.Il12 or Trp2.Combo treated mice; horizontal line represents median, statistical analysis by Kruskal–Wallis with Dunn's tests, $p \leq 0.05$ are shown, the "*" indicates that 1 mouse from the Control group died before performing MRI analysis). **B** Representative images of liver from Control untreated, Ifna.Il12 or Trp2.Combo-treated mice. **C, D** FC analysis of the liver (n = 3, 8, 9 mice/group, for Control untreated, Ifna.Il12 or Trp2.Combo-treated mice; horizontal line represents median, statistical analysis by Mann–Whitney test comparing Ifna.Il12 and Trp2.Combo, $p \leq 0.05$ are shown). **E** IFNγ ELISPOT assay performed on CD8⁺ T cells isolated from the spleen of treated mice (n = 5, mice/

group; horizontal line represents median, statistical analysis by two-sided Mann–Whitney test, $p \leq 0.05$ are shown). **F** Schematics of the antigen prediction pipeline exploited for the identification of neoantigens in the AKTPF LM model. Illustration from BioRender. **G** Schematic of the TA33 LV. **H** Schematic of the experiment shown in (**I–M**). Illustration from BioRender. **I–M** Treatment of mice bearing established AKTPF LM with TA33 of TA33.Combo after 7 days post tumor challenge (TA33 1 × 10⁷ TU/mouse, TA33.Combo total dose 1.2 × 10⁸ TU/mouse). **I** Quantification of LM volume by MRI at day 27 post tumor injection (n = 6, 6, 9 mice/group, for Control untreated, TA33 or TA33.Combo treated mice; horizontal line represents median, statistical analysis by Kruskal–Wallis with Dunn's tests, $p \leq 0.05$ are shown). **J, K** FC analysis of the liver (n = 6, 6, 8 mice/group, for Control untreated, TA33 or TA33.Combo treated mice; statistics as in I). **L, M** IFNγ ELISPOT assay was performed on CD8⁺ T cells isolated from the spleen of the indicated mice (n = 3, 3 mice/group).

By employing TCR tracking analysis, we identified CD8⁺ T cell clonotypes expanded in the livers and LM of both TA33 and TA33.Combo-treated mice (Fig. 6A). Interestingly, expanded TCR clonotypes were mostly shared between the liver and LM. Of note, CD8⁺ T cells bearing expanded and shared TCR clonotypes clustered with CD8⁺ T effector cells in the UMAP plot, representing putative treatment-induced tumor-specific CD8⁺ T cells (Fig. 6B). We observed substantial phenotypic differences in these cells between the TA33 and TA33.Combo groups. Putative tumor-specific CD8⁺ T cells in the TA33.Combo group displayed enhanced expression of genes associated with IFNα and IFNγ signaling, proliferation, and immune activation in both liver and LM compared to the TA33 group (Fig. 6C, D, Supplementary Fig. 12A). Conversely, genes associated with exhaustion and TEX phenotype were downregulated.

In accordance with our previous findings in the MC38.OVA LM model, CD4⁺ T cells in the Combo groups upregulated the expression of genes associated with IFNα and IFNγ signaling as well as with Th1 skewing in both liver and LM (Supplementary Fig. 12B–D). Clonotype tracking of CD4⁺ T cells revealed clonal expansion and sharing between CD4⁺ T cells in the liver and LM, preferentially in the mice treated with TA33.Combo (Supplementary Fig. 12E). In accordance with our previous findings on the MC38-OVA model, CD4⁺ T cells expressing shared clonotypes between liver and LM were enriched in the IFNγ CD4 cluster (Fig. 6B and Supplementary Fig. 12F).

It has been reported that in single-cell RNA sequencing, capture of TCR genes might lead to biased cluster identification, especially when highly expanded T cell clones are present[36]. We repeated our entire analysis, including clustering and cell type annotation after removal of TCR genes, and observed that the exclusion of TCR genes does not substantially affect clustering and cell type identification in our dataset (Supplementary Fig. 13A–D).

To investigate the impact of TA33.Combo treatment on cell-to-cell communication networks within the TME, we performed MultiNicheNet analysis, a computational method to infer active ligand-receptor pairs based on prior knowledge of signaling and gene regulatory networks[37]. In control TA33-treated mice, the top-scoring interactions were among TAMs as well as between TAMs and CD4⁺ T cells (Supplementary data 5). Ligands participating in these interactions are known immunosuppressive molecules (e.g., *Tgfb1*, *Il1b*), pro-angiogenic factors (e.g., *Vegfa*), or other genes associated with tumor progression (e.g., *Fn1*, *Apoe*, *Apoc2*). On the contrary, in the TA33.Combo-treated mice, we observed a rewiring of these interactions, which were predominantly between TAMs/DCs, as senders, and T cells, as receivers. TAMs and DCs were predicted to interact with T cells predominantly through molecules involved in antigen presentation (e.g., *H2.T24, H2.T23, H2.T22, H2.Q6, H2.Q4, H2.M3, H2.D1*) or immune activation (e.g., *Il15, Tnf, Ifng, Il18*). In the TA33 group, we observed that the *Tgfb:Tgfbr2* interaction *was* among the top-scoring interactions between TAMs and CD8⁺ T cells (Fig. 6E). This interaction

is well described to suppress T cell proliferation and effector function[38]. Conversely, in the TA33.Combo group, TAMs, and CD8⁺ T cells were predicted to interact preferentially through molecules involved in antigen presentation and killer cell lectin-like receptor family molecules (e.g., *Klrc1, Klrc2, Klre1*). These receptors have been associated with reduced exhaustion and apoptosis in activated immune cells[39]. Moreover, TAMs/DCs in the TA33.Combo-treated mice may support CD8⁺ T cell proliferation, survival, and effector function through production of IL-15 and activation of IL-2Rb[40]. Additionally, in the TA33.Combo-treated mice, TAMs may inhibit T cell functions through the *Cd274* (PDL1):*Pdcd1* (PD1) axis, which might represent a potential actionable target pathway to enhance the therapeutic effects of the treatment.

To further investigate how TA33.Combo-rewired cellular interactions within the TME, we performed a spatial transcriptomics analysis focused on dissecting the spatial distribution and coordinated expression of selected transcripts in LM upon treatment (Supplementary Fig. 14A, B). TA33.Combo treatment increased the colocalization between transcripts associated with cancer cell identity (e.g., *S100a6, Klf4, Anxa2, Cdkn2a*) and transcripts related to antigen presentation (e.g., *Ciita, H2-Dmb1*), T cell identity (e.g., *Trac*) and T cell activation (e.g., *Sell, Il2rb, Il12rb1*) compared to TA33 treatment (Fig. 6F, supplementary Fig. 14C, D, Supplementary data 6). Transcripts belonging to T cells (e.g., *Cd3e, Trac*) displayed a higher colocalization with transcripts associated with antigen presentation (e.g., *H2-Aa, H2-Ab1*) in the TA33.Combo compared to the TA33 group, suggesting close localization between APCs and T cells. Moreover, in TA33.Combo-treated mice, we observed reduced co-localization of cancer cell identity transcripts with protumoral transcripts (e.g., *Tgfb1, Vegfa, Pdgfb*). In line with MultiNicheNet inferred interactions, *Cd274* colocalized with transcripts related to T cell identity, indicating PDL1:PD1 axis as a potential actionable target to further enhance the therapeutic effects of the treatment.

Overall, these data show that TA delivery alone can expand putative tumor-specific T cells that infiltrate immunosuppressive LM, however the coordinated expression of IL-12 and IFNα is crucial for reprogramming APC functions in both the liver and the TME. TA33.Combo treatment rewires cell-to-cell communication networks, reduces immunosuppressive interactions, and increases antigen presentation and immune cell activation within the TME, supporting the deployment of an effective anti-tumor immune response.

### TA, IFNα and IL-12 LVs restore response to immunotherapy by expanding TA-specific PEX CD8⁺ T cells

We identified 22 genes that are consistently upregulated in tumor-infiltrating T cells upon simultaneous expression of IFNα and IL-12 in both MC38 and AKTPF LMs, hereon termed IFN/IL12-induced T cell (IIT) signature (Supplementary data 7). The IIT signature comprise genes associated with CD8⁺ T cell effector functions (*GZMA, GZMB*), T

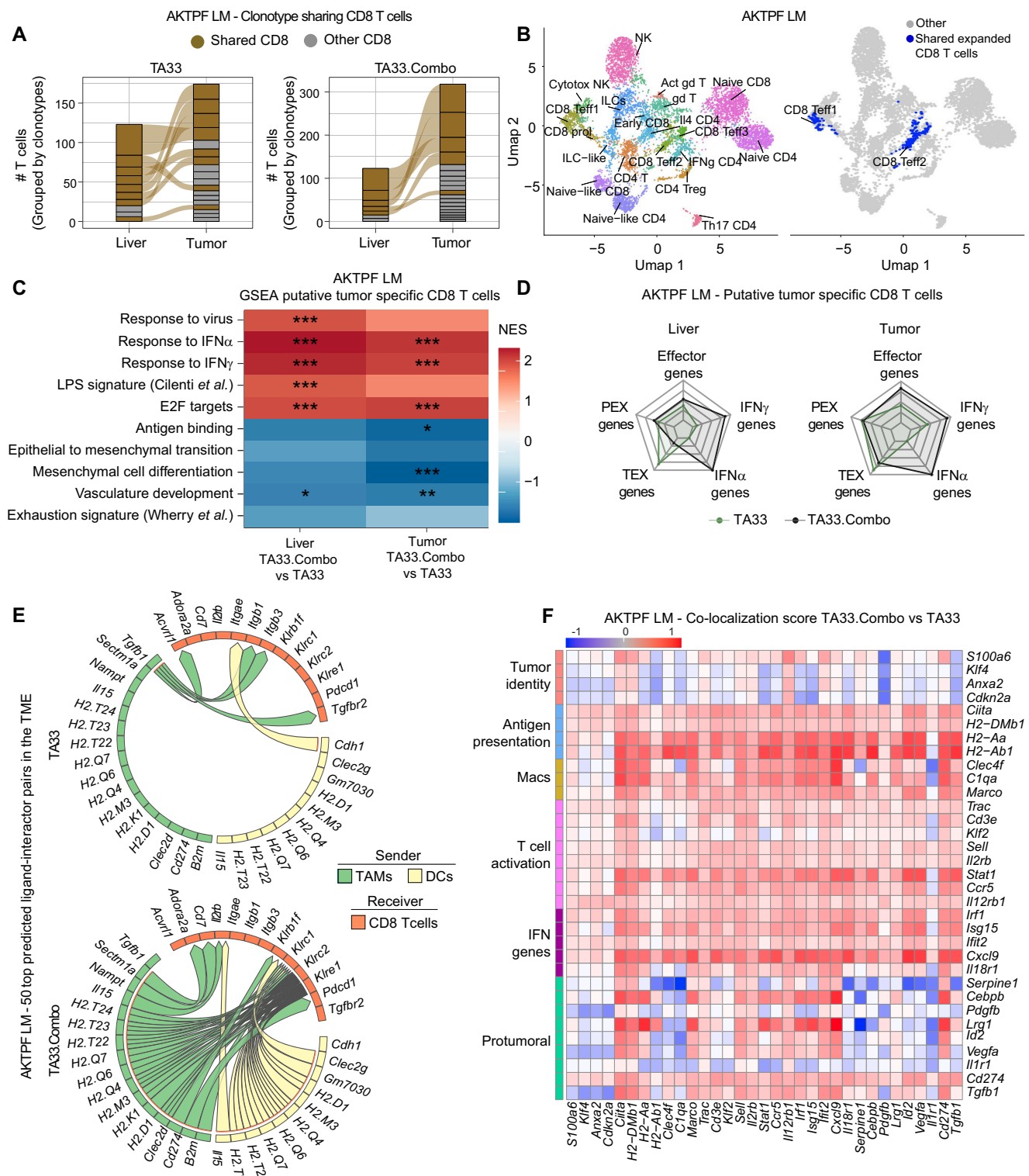

**Fig. 6 | IFNα and IL-12 rewire immune circuits within the metastatic micro-environment enabling TA-specific T cell activity. A** Clonotype sharing between liver CD8[+] T cells, grouped by TCR clonotype (*n* = 3 mouse/group). **B** UMAP projection of scRNA-seq of liver T and NK cell subclusters in left panel, in the right panel representation of shared and expanded CD8[+] T cells. **C** GSEA of scRNA-seq data showing NES for selected GO terms calculated based on genes differentially expressed in putative tumor reactive CD8[+] T cells in the indicated comparisons (*n* = 3 mice/group; statistical analysis by an adaptive multi-level split Monte-Carlo scheme; *: *p*adj < 0.05; **: *p*adj < 0.005; ***: *p*adj < 0.0005). **D** Combined gene expression score for genes belonging to the indicated categories in the indicated

tissue in putative tumor-reactive CD8[+] T cells. **E** Top 50 differential ligand-receptor pairs obtained by MultiNichNet analysis of the tumor scRNA-seq, depicted in circos plots. On the top, TA33, on the bottom TA33.Combo. The arrow indicates the direction from sender to receiver cell type, and the color of the arrow indicates the sender cell type that expresses the ligand. **F** Heatmap showing the co-localization score of transcripts detected by MERSCOPE on liver tissue sections, collected from mice treated with TA33 or TA33.Combo LV, as described in Methods. Samples included both healthy liver parenchyma and liver metastases. The co-localization score was calculated by comparing transcript in TA33.Combo vs TA33 treated animals.

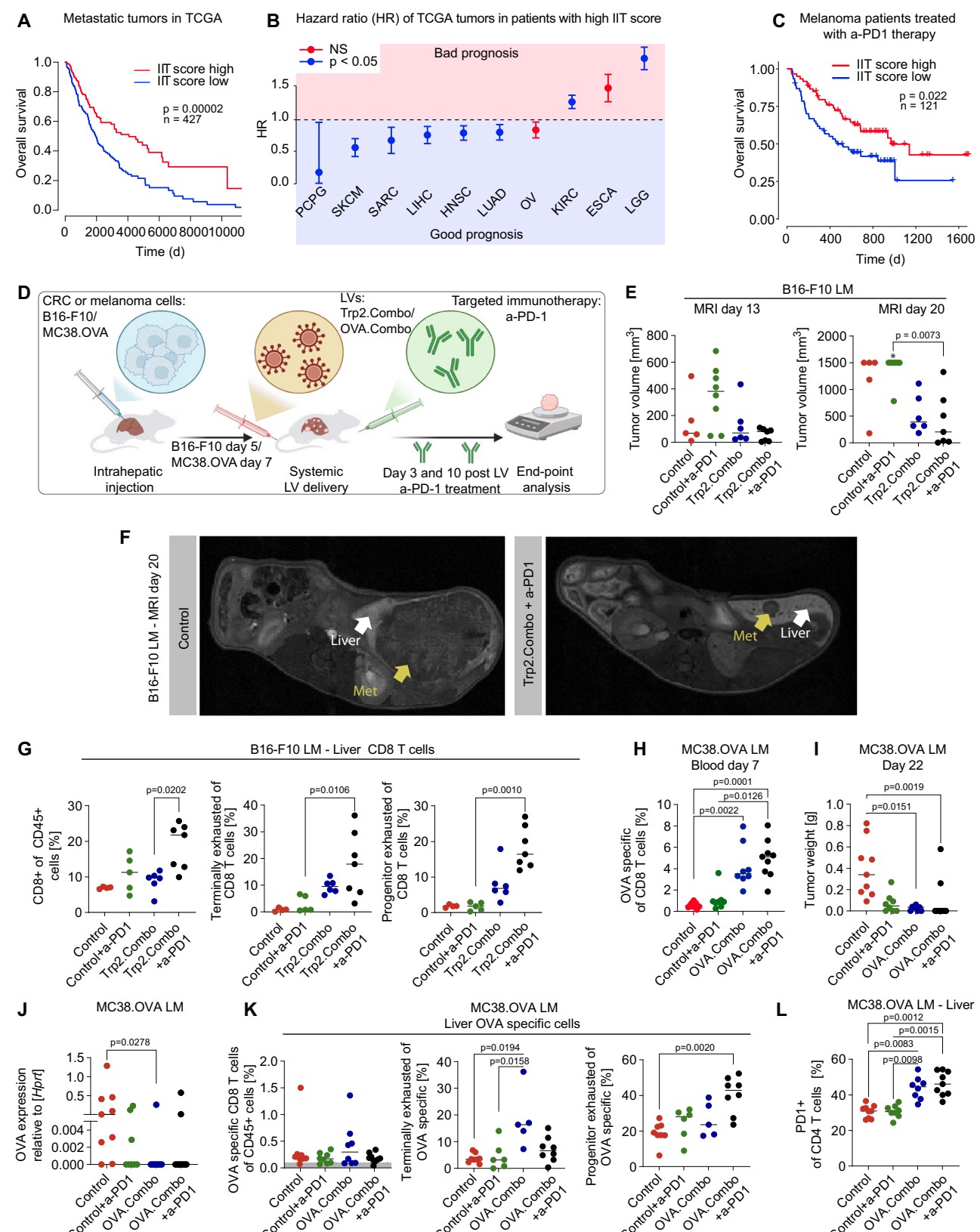

cell memory phenotype development (*KLRG1, CD7, P2RX7*), and T cell migration (*CCR5*), as well as genes with virtually undefined functions in T cells. Remarkably, IIT signature score predicted better survival in patients with distinct types of primary and metastatic tumors (Fig. 7A, B, Supplementary Fig. 15A). Moreover, IIT score positively correlated with improved overall survival in patients treated with anti-

PD1 (a-PD1) immunotherapy (Fig. 7C). Therefore, the IIT signature may be predictive of patient survival and response to immune checkpoint inhibition.

Building on the observation that IIT is associated with response to a-PD1, we investigated whether coordinated delivery of TA, IFNα, and IL-12 could restore response to a-PD1 in mouse tumor models that

**Fig. 7 | TA, IFNα, and IL-12 LVs restore response to immunotherapy by expanding TA-specific PEX CD8⁺ T cells. A** Kaplan–Meier survival curve of metastatic patients with high vs low IIT score (from TCGA, $n = 427$, statistics by Wald test of the Cox regression). **B** Impact of the IIT score on the hazard ratio (HR). Error bars indicate the SE of the regression coefficient. PCPG, ($n = 203$, $p = 0.020$), SKCM ($n = 483$, $p = 0.000011$), SARC ($n = 281$, $p = 0.037$), LIHC ($n = 532$, $p = 0.022$), HNSC ($n = 684$, $p = 0.019$), LUAD ($n = 756$, $p = 0.038$), OV ($n = 441$, p = 0.097), KIRC (n = 1080, p = 0.030), ESCA ($n = 228$, $p = 0.071$), LGG ($n = 548$, $p = 0.000171$). **C** Kaplan–Meier survival curves of metastatic patients treated with a-PD1 and showing high vs low IIT score ($n = 121$). Statistics are calculated as in (**A**). **D** Schematic of the experiment shown in **E–G** and **H–L**. Illustration from BioRender. **E–G** Treatment of mice bearing established B16-F10 LM (Control mice were left untreated, Trp2.Combo total dose of $1.2 \times 10^8$ TU/mouse). **E** LM volume was measured by MRI ($n = 5, 8, 6, 7$ mice/group, for Control untreated, Control + a-PD1, Trp2.Combo or Trp2.Combo + a-PD1 groups; the horizontal line represents median, statistical analysis by Kruskal–Wallis with Dunn's tests, $p \leq 0.05$ are shown. The "*" indicates that 1 mouse died before performing MRI analysis. **F** representative MRI images. **G** FC analysis of the liver ($n = 3, 5, 6, 7$ mice/group, for Control untreated, Control + a-PD1, Trp2.Combo or Trp2.Combo + a-PD1 treated mice; the horizontal line represents median, statistical analysis by Kruskal–Wallis with Dunn's tests, $p \leq 0.05$ are shown). **H–L** Treatment of mice bearing established MC38.OVA LM (Control LV $1.2 \times 10^8$ TU/mouse, OVA.Combo total dose of LV $1.2 \times 10^8$ TU/mouse). **H** FC analysis of the blood ($n = 9, 7, 8, 9$ mice/group, for Control untreated, Control + a-PD1, OVA.Combo or OVA.Combo + a-PD1 treated mice; horizontal line represents median, statistical analysis by Kruskal–Wallis with Dunn's tests, $p \leq 0.05$ are shown). **I** Tumor weight (number of mice and statistic as in **H**). **J** OVA gene expression analysis on residual LM (number of mice and statistics as in H). **K** FC analysis of the liver (number of mice and statistics as in **H**). **L** FC analysis of the liver (number of mice and statistics as in **H**).

normally do not respond to checkpoint inhibition, such as B16-F10 melanoma LM[41]. To this aim, we inoculated B16-F10 cells intrahepatically in C57BL/6 mice to create experimental LMs (Fig. 7D) and then treated the mice with a-PD1, Trp2.Combo or the combination of these two treatments. After LV and a-PD1 delivery, we detected similar levels of IFNα and IL-12 in the plasma of Trp2.Combo and Trp2.Combo + a-PD1-treated mice (Supplementary Fig. 15B). Trp2.Combo + a-PD1 treatment increased the fraction of PD1⁺ CD8⁺ T cells and effector CD62L⁻ CD44⁺ CD8⁺ T cells compared to control untreated or a-PD1-treated mice (Supplementary Fig. 15C). Of note, in Trp2.Combo + a-PD1 treated animals we also observed a significant increase in the activated Ly6c⁺ CD44⁺ CD8⁺ T cells in circulation, which negatively correlated with the tumor volume (Supplementary Fig. 15D). This finding suggests that these cells may comprise putative tumor-specific CD8⁺ T cells.

In agreement with enhanced immune activation in Trp2.Combo- and Trp2.Combo + a-PD1-treated mice, we observed delayed LM growth (Fig. 7E, F). Of note, only Trp2.Combo + a-PD1 treatment led to near-complete LM eradication in 3 out of 7 mice. On the contrary, a-PD1 alone did not exert a beneficial therapeutic effect in this model. The addition of a-PD1 to the Trp2.Combo treatment resulted in an increase in both the overall fraction of liver-infiltrating CD8⁺ T cells and the fraction of PEX CD8⁺ T cells, in agreement with previous studies showing that a-PD1 acts specifically on this subset of CD8⁺ T cells[42,43]. (Fig. 7G). Moreover, in Trp2.Combo and Trp2.Combo + a-PD1-treated mice we observed increased PD1 expression on liver CD4⁺ T cells, as previously observed in the MC38.OVA model (Supplementary Fig. 15E).

In the LM we observed enhanced infiltration of CD8⁺ T cells in both Trp2.Combo and Trp2.Combo + a-PD1 groups compared to controls (Supplementary Fig. 15F). Trp2.Combo enhanced the fraction of PEX CD8⁺ T cells, which were further expanded by combination with a-PD1. CD4⁺ T cells infiltrating LM displayed a Th1-like phenotype in both Trp2.Combo groups, again with a stronger effect with a-PD1 addition (Supplementary Fig. 15G). Interestingly, the Trp2.Combo + a-PD1 resulted in a reduced fraction of CD4⁺ Tregs compared to a-PD1 alone. Overall, these findings suggest that Trp2.Combo treatment restores response to a-PD1 in immune-cold tumors, by favoring the expansion of PEX CD8⁺ T cells.

To better understand the effects of a-PD1 on treatment-induced tumor-specific T cells, we repeated the experiment in mice bearing MC38.OVA LM (Fig. 7D). After LV delivery, we detected similar levels of IFNα and IL-12 in the plasma of OVA.Combo- and OVA.Combo + a-PD1-treated mice (Supplementary Fig. 15H). As expected, we observed an increased fraction of circulating OVA-specific CD8⁺ T cells in OVA.-Combo-treated mice compared to controls (Fig. 7H and Supplementary Fig. 15I). Both OVA.Combo and OVA.Combo + a-PD1 reduced LM growth compared to control and favored eradication of OVA-expressing cancer cells (Fig. 7I, J). Notably, OVA.Combo + a-PD1 led to a stronger effect, with 7 out of 9 mice resulting tumor free at the termination of the experiment. Interestingly, we found expression of OVA in LM from the two mice that did not respond to OVA.Combo + a-PD1, suggesting development of resistance to the treatment.

Both OVA.Combo and OVA.Combo + a-PD1 treatments increased the fraction of OVA-specific CD8⁺ T cells displaying features of PEX, with OVA.Combo + a-PD1, leading to stronger effect (Fig. 7K). On the other hand, OVA.Combo + a-PD1 did not increase the fraction of CD8⁺ T cells displaying features of TEX, in agreement with findings from previous reports[42,43]. Moreover, both OVA.Combo treatments increased the fraction of CD4⁺ T cells expressing PD1 compared to control mice or a-PD1-treated mice, suggesting CD4⁺ T cells activation upon a-PD1 treatment only in the presence of IFNα and IL-12 (Fig. 7L).

Alogether, these results indicate that concurrent delivery of TA, IFNα, and IL-12 activates a genetic program in tumor-infiltrating T cells that may restore the therapeutic effect of a-PD1, providing proof-of-concept of a powerful therapeutic combination to treat cancer patients with an unmet clinical need.

## Discussion

We developed a modular platform to simultaneously drive the expression of TAs in combination with locally sourced immune-activating cytokines in the metastatic liver. We leveraged on recently established LVs, which enable selective expression in liver macrophages in proximity to LM and can accommodate tunable payload combinations. LV-driven TA presentation in both MHC-I and MHC-II prevented growth of TA-expressing cancer cells. However, in presence of established tumors, addition of LVs driving expression of IFNα and IL-12 was necessary to achieve TA specific immunity and enhanced therapeutic efficacy in LM from distinct tumor types. The cytokines enhanced the expression of genes involved in antigen presentation in APCs, increased the fraction of treatment-induced TA specific PEX CD8⁺ T cells independently of cognate MHC-II TA delivery and CD4⁺ T cell activation. Addition of PD1 blocking moAbs resulted in further expansion of TA specific PEX T cells achieving tumor eradication in most treated mice.

Liver macrophages possess the ability to prime and activate CD8⁺ T cells as evidenced in studies of hepatotropic viral infections in mouse models[44]. Liver APCs, including KCs, have been exploited to generate liver-directed vaccines, which promote immune activation against tumor or viral antigens in the liver[45–47]. Exploiting liver macrophages to drive TA expression and cytokine production offers several advantages over conventional therapeutic vaccines targeting professional APCs in lymph nodes or other immunological compartments. Expression of immune-activating cytokines from within the liver enables robust reprogramming of the immunosuppressive liver microenvironment, supporting local and systemic immune responses. For example, liver cell populations have been exploited to produce immune activating cytokines such as IFNα or IL-2[48,49]. Moreover, liver macrophages are an abundant population of APCs, which due to their location within the liver sinusoids preferentially interact with

circulating and liver resident or LM-infiltrating T cells. Local activation of TA specific T cells in the liver may facilitate their localization within LM and protect the liver from further tumor invasion. On a more speculative level, enforcing TA presentation and co-stimulation by liver resident and tumor-infiltrating APCs may achieve more effective priming and foster antigen spreading.

The success of a cancer vaccine hinges on the selection of the right TA. In this study we explored the platform's adaptability by employing various TAs that differ in expression pattern and immunogenicity. We successfully generated TA specific CD8[+] T cells against a surrogate antigen (OVA), a TAA (Trp-2) and tumor-specific neoantigens identified in a mouse model of LM (TA33). Prophylactic TA delivery resulted in protective immunity against LMs and the generation of TA-specific PEX CD8[+] T cells. Moreover, prophylactic delivery of TA in combination with cytokines protected mice from the engraftment of intrasplenically injected AKTPF cells, a model that recapitulates key features of the metastasis process.

The generation of protective TA reactive T cell depended on antigen load in our study. Although counterintuitive, we observed that higher dose of TA lead to a less robust anti-tumoral effect. It has been observed that increased doses of TA may be beneficial in some settings[50,51]. However, it also has been observed that sustained and robust expression of antigens can lead to defective T cell activation, hence reduced tumor control[52–54]. It is possible that in our setting of LV-based liver-directed delivery of TA, both the extent of cells expressing the antigen, the level of antigen presentation per cell, the immunogenicity of the TA, and the immune features of the liver may contribute to the observed behavior.

Another relevant finding lays in the necessity of an MHC-II moiety in the delivered TA when delivering it in prophylactic settings and without cytokines. This finding agrees with reports that indicate that the lack of CD4[+] T cell help results in the generation of dysfunctional CD8[+] T cells[55], and it supports the contention that to achieve effective anti-tumor immunity, TA-specific CD8[+] T cells with features of PEX must be maintained[19,56]. Of note, to generate protective immunity upon TA LV delivery, the MHC-II moiety of the liver macrophage-directed TA LV did not need to be tumor-specific, in accordance with recently reported universal MHC-II epitopes, which enhanced the therapeutic efficacy of tumor vaccines[57]. Of note, in presence of gene-based delivery of IFNα and IL-12, the MHC-II moiety in the TA LV as well as the function of CD4[+] T cells were dispensable to enable CD8[+] T cell effector function against established LMs. These results highlight how co-delivery of IFNα and IL-12 may potentiate cancer vaccines and simplify their design in translational applications. Indeed, predicting epitopes for binding with MHC-I is a more straightforward task compared to MHC-II[58,59].

The different therapeutic outcome of LV-based TA delivery observed in the prophylactic compared to therapeutic setting could have several explanations. First, in a prophylactic setting, TA presentation by engineered liver macrophages primes naïve CD8[+] T cells, whereas in a therapeutic setting, liver macrophages present the antigen to T cells, which were already primed in the tumor or in tumor-draining lymph nodes. This might suggest that TA delivery is sufficient to properly activate naïve CD8[+] T cells, but to a lesser extent to reprogram activated ones. Second, in the therapeutic setting, the presence of the tumor might have an impact on the phenotype of liver resident macrophages, reducing their ability to present antigens or to co-stimulate T cells. Last, in a therapeutic setting, TA specific CD8[+] T cells need to infiltrate a well-established tumor, which might suppress T cell functions and impair their infiltration[60]. Conversely, in the prophylactic setting CD8[+] T cells prevent the engraftment or growth of a low number of cancer cells. Whereas a more thorough investigation may be needed to discern the differences in phenotype between liver macrophages and TA specific CD8[+] T cells during prophylactic or therapeutic interventions, our results show that reprogramming of the immune cells in the liver by immune-activating cytokines is necessary to promote effective immune responses against established LM.

Simultaneous delivery of a TA in combination with IFNα and IL-12 showed additive effects resulting in superior therapeutic efficacy compared to TA and cytokines alone. Our results are in line with previous studies, including our own, reporting that IFNα enhances the expression of MHC-I-associated genes in APCs and cancer cells and reduces the exhaustion of T cells[15], and that IL-12 activates both macrophages, DCs and T cells in vivo[61]. In previous studies knockout of IFNα receptor (IFNAR) in T cells did not alter the therapeutic efficacy of gene-based delivery of IFNα[15], suggesting that the effect on T cells observed upon IFNα enforced expression may be mediated by the reprogramming of TME, including APCs. On the contrary, IL-12 has been shown to drive T cell activation and proliferation directly as well as by inducing IFNγ released by both APCs and T cells[62]. IL-12 is known to promote IFNγ production in T cells, APC licensing and ultimately CD8[+] T cell cytotoxic activity against cancer cells[62]. Overall, while it is clear that we were able to achieve additive effect by combining IFNα and IL-12 delivery, is not possible to determine whether the repro-graming activity of combinatorial IFNα and IL-12 treatment is a direct consequence of engagement of cognate receptors on T cells and APCs or an indirect effect resulting from APC activation and TME reprogramming.

OVA is a highly immunogenic surrogate TA, which enables development of spontaneous anti-OVA immune responses. On the contrary, TAAs are often weakly immunogenic because they are also expressed, albeit to a lesser extent, by healthy tissues. Thus, T cells bearing TCR clonotypes that recognize them are mostly eliminated during thymic selection. This tolerogenic mechanism prevents T cell responses against self-antigens, potentially explaining the limited response against the TAA observed in the B16-F10 model. It has been previously shown that IFNα and IL-12, promote therapeutic responses in distinct tumor types by reprogramming immune functions[15,63–65]. However, combining IL-12 and IFNα with TAA delivery led to improved anti-tumor immunity compared to cytokines alone, highlighting the importance of exogenous TA delivery for robust presentation which, in the presence of immune activating cytokines, may effectively activate T cells with lower affinity for the TA or present at very low frequency.

Overall, our study provides the foundation for a generation of cancer vaccines built by a stackable assembly of liver macrophage targeting LVs that can accommodate tunable combinations of immune stimulatory cytokines and TAs. Further preclinical studies aimed to establish the safety, biodistribution and pharmacodynamics of this platform in suitable animal models should allow moving forward to address the severe unmet clinical need of patients with LM.

Our study suffers from the inherent limitations of using tumor mouse models. To overcome some of the limitations of using a single mouse model, we employed distinct mouse models of LMs, accounting for differences in terms of primary origin and immunogenicity of the tumors.

We also need to consider the potential risk of toxic side effects induced by cytokine delivery. Indeed, while cytokines play a pivotal role in modulating immune responses, their administration may carry the risk of inducing toxic side effects. The systemic effects of inflammatory cytokines can impact various physiological processes beyond the intended immune modulation, and this aspect warrants careful consideration. Our strategy enables cytokine delivery from within the tissue, bypassing vascular barriers, likely resulting in higher and more constant levels of the cytokines within the liver and lower systemic exposure than obtained after conventional administration of recombinant proteins. However, cytokines were still detectable in the circulation in the range of hundreds pg/mL, which may trigger inflammatory and toxic responses in other tissues, especially in the case of IL-12. Recent studies have shown that by improving IL-12 on-tumor targeting it is possible to reduce adverse side effects[14,66–68]. We investigated safety

of our strategy by monitoring mouse weight, liver transaminases, circulating cytokines, blood parameters and tissue histopathology, confirming the safety and tolerability of our combination treatment. However, refinements in the dosage of LVs might be needed to obtain more controlled and reproducible levels of the cytokines. Of note, emerging toxic side effects might be mitigated by delivering anti-IFNα and anti-IL-12 antibodies, as well as by employing liver macrophage-depleting strategies such as bisphosphonates, which have been approved for clinical use. Another study has shown that IL-12-induced toxicity may be mitigated by employing anti-TNFα antibodies[26]. Of note, we observed that the cytokine levels dropped over time, likely due to the natural turnover of liver resident macrophages and to the counter selection of transduced cells as observed previously[15]. Importantly, activation of TA specific T cells results in the killing of the liver macrophages presenting the TA[22], thus reducing over time the fraction of transduced cells as well as the expression of cytokines.

Another concern possibly raised by our study is associated with the employment of integrating viral vectors such as LVs. However, it is worth noticing that, in the last years, a large number of patients have been treated with LV-engineered cell products including LV-transduced hematopoietic stem cell (HSCs) for the treatment of hereditary genetic diseases as well as CAR LV-transduced T cells for the treatment of distinct types of tumors, many of which have become registered drugs for the market. These safe and successful applications of LVs support the translatability potential of our study to patients with LM.

## Methods

### Plasmid design

The MRC1.empty. miRT (Control) transfer vector, as well as the dOVA, IFNα and IL-12 transfer plasmid were already present in the lab[15]. The Ii.OVA transfer plasmid was a kind gift from Andrea Annoni, while Ii.SIINFEKL was obtained by PCR amplification with specific oligos indicated in Supplementary table 1 using Phusion High-Fidelity DNA Polymerase (Thermo Scientific/ F-530XL). The TB fragment was obtained by annealing synthetic oligos indicated in supplementary Table 1 (A85B_S and A85B_AS). The TRP-2 fragment was obtained by PCR on cDNA obtained by retrotrascription of RNA extracted from B16-F10 cells, using oligos indicated in supplementary table 1. The AKPTF TA33 DNA sequence was produced by Twinhelix. The PCR products and the TA sequences were purified with NucleoSpin Gel and PCR Clean-up kit (Macherey-Nagel/740609.50) processed by enzymatic digestion with MluI and NheI, and cloned downstream of the invariant chain of the Ii.OVA transfer vector.

Ligation products for all constructs were transformed in bacteria by adding about 100 ng of ligation product to 50 μL of competent bacteria (Top10 cells- Invitrogen). The retransformation mix was carefully mixed, incubated for 30 minutes on ice, transferred to 42 °C for 30 seconds followed by two additional minutes incubation on ice. 500 μL of Luria-Bertani (LB) medium was add and incubated for 30 min at 37 °C. Bacteria were plated onto LB agar plates containing 100 μg/mL ampicillin and incubated overnight at 37 °C. Afterwards, single colonies were picked and transferred into 3 mL LB medium containing 100 μg/mL of ampicillin and incubated overnight at 37 °C shaking at 150 rpm. DNA was extracted from 2 mL of the bacteria culture using the Wizard Plus SV Minipreps DNA Purification System kit (Promega/A1330). Plasmids derived from different clones were screened by analytical digest as well as Sanger sequencing to confirm plasmid sequence. To further amplify the plasmids, the remaining 1 mL of the bacteria liquid culture from clones containing plasmids with the correct sequence was used to inoculate 500 mL of LB medium containing 100 μg/mL of carbenicillin, which were subsequently left to grow overnight at 37 °C shaking at 150 rpm. DNA extraction was performed using Nucleobond Xtra Maxi EF (Macherey-Nagel/740424.50), and the final plasmid was resuspended in endotoxin-free TE.

### LV production and titration

In this study, third-generation self-inactivating VSV-G pseudotyped LVs were used. LV stocks were produced on a laboratory scale as described previously[69]. In brief, LVs were produced in HEK293T, which were transfected with distinct transfer plasmid (respectively carrying TA, IFNα or IL-12) along with third-generation packaging plasmids, pGag-Pol, pRSV-REV, pAdVAntage as well as an envelope protein plasmid encoding VSV.G. After 14 h from the transfection, media was replaced, and after additional 30 h, the LV-containing supernatant was harvested, filtered and ultracentrifuged. The pellet containing the LV particles was then resuspended in fresh PBS and stored at −80 °C. The produced LVs were then titrated on HEK293T cells to determine the TU/mL of the produced vector, as previously described[69].

### Vector copy number determination

Genomic DNA was extracted from cell culture samples using the Maxwell RSC 48 Instrument (Promega/AS8500) with Maxwell RSC Cultured Cells DNA Kit (Promega/AS1620). Genomic DNA from whole tissue samples was extracted by using the DNeasy Blood and Tissue Kit (Qiagen/69504) according to manufacturer's instruction. LV copy number was determined using a QX200 Droplet Digital PCR System (Biorad) apparatus, and the digital droplet PCR was performed according to manufacturer's instructions.

Briefly, for each sample a reaction was prepared containing ddPCR Supermix for Probes (No dUTP) (Biorad/1863024), 10-50 ng of genomic DNA, primers at a final concentration of 900 nM, and the detection probes at 250 nM. Primers and probes for the detection of HIV genomes and Sema3a gene, employed as normalizer for murine samples, are reported in supplementary table 1. As normalizer for human samples, a commercially available GAPDH expression assay was used (Invitrogen/Hs00894322_cn).

After droplet generation, the plate was sealed and amplified in a thermal cycler with following protocol:

| | | |
|---|---|---|
| 95 °C | 10 min | 1× |
| 94 °C | 30 sec | 40× |
| 60 °C | 1 min | |
| 98 °C | 10 min | 1× |
| 4 °C | ∞ | 1× |

Amplified droplets were acquired using the BioRad QX200 Droplet Reader and analyzed by using the QuantaSoft software (Biorad). LV copies per genome were calculated by the formula:

$$\text{LV copies per genome} = \text{concentration HIV}/\text{concentration Normalizer}*2$$

(1)

### Gene expression assays

For gene expression analysis, RNA was extracted from pelleted cells using the miRNeasy Micro Kit (Qiagen/217084). Retrotranscription was performed according to manufacturer's instruction using the SuperScript IV VILO (Invitrogen/11756050). We used 0,25-2,5 ng of cDNA as input for the gene expression analysis. TaqMan Gene Expression Assays from Invitrogen are listed in supplementary table 2. Data acquisition using ddPCR and analysis were performed as described above.

### HEK293T, MC38.OVA, and B16-F10 cells

Human embryonic kidney 293T (HEK293T) cells were employed to produce and titer LVs, they were purchased from ATCC and their authenticity is supported by their capability to produce high titer LV

stocks. MC38.OVA and B16F10 that were used to mimic LM upon liver implantation were already present in the lab and were previously obtained from Amgen[15].

All three cell lines were cultured in adherence at 37 °C in IMDM medium (Corning/10-016-CV) supplemented with 10% FBS (HyClone/SH30066.03), 100 IU/mL penicillin, and 100 μg/mL streptomycin. Cells were split three times a week 1:5–1:10 by removing culture medium, washing cells with Phosphate Buffered Saline (PBS; Corning/21-031-CVR), and detaching them with a solution of 0.05% trypsin and EDTA (4 mM) in PBS (ATV). Cells were then resuspended in a fresh medium and transferred into a new plate.

### AKTPF cells
AKTPF cells were employed to mimic CRC LM upon intrasplenic implantation. They were obtained from the laboratory of Masanobu Oshima[33].

Cells were cultured in adherence at 37 °C in Advanced Dulbecco's Modified Eagle Medium (DMEM)/F12 medium (Gibco/12634-010) supplemented with 10% FBS (HyClone/SH30066.03), 2% GlutaMAX supplement (Thermo Scientific/35050061), 100 IU/mL penicillin, and 100 μg/mL streptomycin. Cells were split three times a week 1:8–1:10 by removing culture medium, washing cells with PBS (Corning/21-031-CVR), and detaching them with a solution of 0.05% trypsin and EDTA (4 mM) in PBS (ATV). Trypsin solution was inactivated by adding fresh medium, cells were pelleted 5 min 200 g 4 °C and resuspended in fresh medium before being transferred into a new plate.

### Mouse procedures
In this study, we employed female C57BL/6 mice purchased from Charles River Laboratory. All experiments and procedures were performed according to protocols approved by the Institutional Animal Care and Use Committee (IACUC) at San Raffaele Hospital animal facilities (IACUC number: 1098, 1227, 1383, 1462) and authorized by the Italian Ministry of Health and local authorities according to the Italian law. Mice were used between 5 and 10 weeks of age and were maintained in specific pathogen-free animal research facilities with a 12/12 h dark/light cycle and standardized temperature ($22 \pm 2$ °C) and humidity ($55 \pm 5\%$). To prevent unnecessary suffering, health parameters were monitored, such as loss of weight, breathing, mobility, stress, manipulation response, fur status, and tumor growth. The maximal tumor burden was set at 1000 mm³, beyond which mice were sacrificed. If MRI measurements indicated a tumor volume exceeding this limit, the affected mice were sacrificed immediately in accordance with ethical guidelines.

### Intravenous injection of LV
LVs were diluted in PBS to obtain the desired TU/mouse in a volume of 200 μL. Before injection, mice were warmed under an infrared/red-light lamp and the LVs were delivered in the tail vein. For the injection of multiple LVs (e.g., OVA.Combo, Trp2.Combo, TA33.Combo), the different LV preparations were mixed and diluted in PBS to achieve the desire TU/mouse. In prophylactic experiments, LVs were injected 14 days prior to tumor challenge, while in therapeutic experiments, LVs were injected 3-, 5- or 7 days post-tumor placement. In general, LVs carrying the TA or IL-12, were injected at a dose of $10^7$ TU/mouse while the LV carrying IFNα was administered at $10^8$ TU/mouse. Specific timing and dosage of the LV injection are indicated either in the main text, in the figures, or in the figure legends.

### MC38.OVA/B16-F10 tumor implant
We delivered MC38.OVA subcutaneously by injecting $10^6$ MC38.OVA cells into the flank of mice in a volume of 50 μL of Matrigel (BD Biosciences) diluted 1:2 in PBS. Tumor growth was monitored by measuring the dimensions (larger diameter, *x*, and lower diameter, *y*) of the subcutaneous lesions using a caliper. Tumor volume was calculated

with the formula:

$$\text{Volume} = \text{diameter}(x)^2 * \text{diameter}(y)/2 \qquad (2)$$

For intrahepatic injection of MC38.OVA or B16-F10 cells, mouse fur was removed from the abdominal area of the mice by shaving. Immediately prior to surgery, mice were injected with 50 μL carprofen (2.5 mg/mL) for pain management. Isoflurane (Iso-Vet 104331020) at a concentration of 3% in flow of oxygen at 1.5 L/min was used to anesthetize the mice during surgery. We injected 100,000 MC38.OVA or 20,000 B16-F10 cells/mouse in 5 μL/mouse of PBS with a Hamilton syringe, in the left liver lobe. Following surgery, mice were subjected to antibiotic treatment for one week by adding Baytril (Bayer) at a concentration of 0.5 mg/mL to the drinking water. LM growth was measured by tumor weight (i.e., by dissecting the LM upon experiment termination and measuring its weight on a precision digital bench scale).

### AKTPF tumor implant
We delivered AKTPF cells by intrasplenic injection. Briefly, mouse fur was removed at the left upper flank of the mice by shaving. Immediately prior to surgery, mice were injected with 50 μL carprofen (2.5 mg/mL) for pain management. During surgery, mice were anesthetized as previously mentioned, and 30,000 AKTPF cells/mouse were resuspended in 50 μL/mouse Geltrex (Life Tehnologies/A1413301) and carefully injected into the spleen using a precooled 0.5 ml syringe with 28 G needle. The peritoneum wall was sutured by using adsorbable stitches while the skin was closed by applying stainless steel wound clips. Following surgery, mice were subjected to antibiotic treatment. LM growth was measured by using MRI. To perform splenectomy, mice were anesthetized using isoflurane to ensure proper sedation. Following sterilization of the surgical area, a small incision was made on the left flank using sterile surgical instruments to expose the spleen. The splenic circulation was carefully ligated using a surgical suture to prevent hemorrhage, and the spleen was removed. The incision was then closed using appropriate sutures and wound clips.

### Blood collection and analysis
Blood was withdrawn from the retroorbital vein plexus and collected in Microvette with EDTA (Sarstedt/NC9990563). Hemocytometer analysis was performed on whole blood by using the ProCyte DXTM (IDEXX). For the collection of plasma, blood was centrifuged at 850 g for 10 minutes at room temperature, and precipitated red and white blood cells were discarded. Quantification of IFNα content in the blood was performed on plasma using the Mouse IFN Alpha All Subtypes ELISA KIT High Sensitivity (pbl Assay Science) according to manufacturer's instruction. Mouse IL-12 p70 was measured with Legend MAX™ Mouse IL-12 (p70) ELISA Kit (Biolegend), according to manufacturer's instruction. Bio-Plex Pro Mouse Cytokine 23-plex Assay #M60009RDPD (Biorad) was performed according to manufacturer's instruction. Negative values were considered 0 in the analysis.

For the assessment of transaminases in the serum, ALT (Instrumentation Laboratory) and AST (Instrumentation Laboratory) quantification kits were used with an International Federation of Clinical Chemistry and Laboratory Medicine–optimized kinetic ultraviolet (UV) method in an ILab Aries chemical analyzer (Instrumentation Laboratory). In parallel, SeraChem Control Level 1 and Level 2 (#0018162412 and #0018162512) were analyzed as quality control.

### Magnetic resonance imaging
A 7-Tesla preclinical scanner (Bruker, BioSpec 70/30 USR, Paravision 6.0.1), equipped with 450/675 mT/m gradients (slew-rate: 3400–4500 T/m/s; rise-time 140 ms) and a circular polarized mouse body volume coil with an inner diameter of 40 mm was used. During acquisition, mice were kept under anesthesia by inhaling isoflurane

(Iso-Vet/ 104331020) at a concentration of 3% inflow of oxygen at 1.5 L/ min under a dedicated temperature control apparatus to prevent hypothermia. The breathing rate and the body temperature were continuously monitored (SA Instruments, Inc., Stony Brook, NY, USA). To aid liver lesion visualization, we used a hepatocyte-specific contrast agent, the Gd-EOB-DTPA (Primovist, Bayer Schering Pharma), at 0.05 mmol/g of body weight. Axial fat-saturated T2-weighted images (RARE-T2, Rapid Acquisition with Relaxation Enhancement, TR = 3000 ms, TE = 40 ms, voxel size = 0.125 3 0.100 3 0.8 mm, averages = 4,) and axial fat-saturated T1-weighted sequences (RARE-T1: TR = 540 ms, TE = 7.2 ms, voxel size = 0.125 3 0.100 3 0.8 mm, averages = 4) were acquired during the hepatobiliary phase of Gd- EOB-DTPA enhancement (10 minutes after administration). Volume measurement was performed by using the Medical Image Processing, Analysis, and Visualization software.

## Monoclonal antibody injection

Monoclonal antibodies were injected at a concentration of 0.2 mg/ mouse, by an intraperitoneal injection in 100 μL, diluted in PBS. a-PD1 mabs were injected after 3 and 10 day post LV injection. a-CD4, a-CD8, and a-NK1.1 were injected one day before LV treatment and then twice a week until experiment termination.

The antibodies used were In-vivoMAB anti-mouse PD-1 (BioXCell, catalog number:BE0146), In-vivoMAB anti-mouse CD4 (BioXCell, catalog number:BE0003-1), In-vivoMAB anti-mouse CD8a (BioXCell, catalog number:BE0004-1), In-vivoMAB anti-mouse NK 1.1 (BioXCell, catalog number:BE0036).

## Sacrifice and necropsy

For endpoint analysis, mice were euthanized by cervical dislocation. The liver was perfused by injecting 10 mL of PBS containing 5 mM of UltraPure EDTA pH 8 (Invitrogen/ 15575020) through the inferior vena cava and cutting the portal vein to allow exiting of the solution containing most circulating blood cells from the liver. When FC analysis, but not immunofluorescence (IF) analysis, was performed, 10 mL of IMDM (Corning) containing 0.35 mg/mL collagenase (Sigma-Aldrich) was injected through the inferior vena cava. All organs were collected and immediately stored on ice (for FC and IF) or dry ice (for DNA/RNA extraction). The spleen was collected in a sterile condition and then processed for splenocyte freezing. In brief, the spleen was smashed in a petri dish, washed with IMDM (Corning), and the pellet was resuspended in 1 mL IMDM + DMSO and frozen.

## IFNγ ELISpot

Multiscreen filter plates (Millipore-Merck) were coated overnight 4 °C with purified anti-mouse IFNγ mAb (clone R46A2 5 μg/mL 50 μL/well, BD-Pharmingen) and blocked with PBS 1% BSA for 2 hours at 37 °C. Plates were equilibrated with culture medium for 10 minutes at room temperature before seeding cells. Splenic CD8 + T cell were negatively selected by magnetic beads sorting kit according to manufacturer's recommendation (Miltenyi), plated ($1 \times 10^5$ cells/well) at least in duplicates in RPMI 1640 (Lonza) supplemented with 10% fetal bovine serum (FBS) (Euroclone), 100 U/mL penicillin/streptomycin (Lonza), 2 mM L-glutamine (Lonza), 0.1 mM Minimum Essential Medium Non-Essential Amino Acids (MEM NEAA) (Gibco), 1 mM Sodium Pyruvate (Gibco), 50 nM 2-Mercaptoethanol (Gibco). Antigenic stimulation was provided by co-culture at ratio 1:1 with syngeneic irradiated (60 Gy) EL4 cell line pulsed with the indicated peptide or peptide-pool, or not with wt irradiated EL4 cell line in the presence of 50 U/mL IL-2 (Proleukin, Chiron) for 42 hours at 37 °C, 5% CO2. As positive control of IFNγ release, splenic CD8$^+$ T cells were polyclonally stimulated with 10 ng/ml of phorbol myristate acetate (PMA) and 1 μM calcium ionomycin (Sigma-Aldrich). At the end of the culture detection, biotinylated anti-mouse IFNγ mAb (XMG1.2 1 μg/mL 50 μL/well, BD-Pharmingen) was added and incubated for 2 hours at room

temperature. Avidin-POD solution (Roche, 1:5,000, 50 μL/well) was then added and incubated for 1 hour at room temperature. IFNγ-spots were developed by AEC solution (Sigma-Aldrich) at room temperature for 15 minutes in the dark. Plate images were acquired, and spots were counted by Immunospot S6-Ultra (Cellular Technology Limited). Data are reported as number of IFNγ-spot forming unit (SFU) in $10^6$ CD8$^+$ T cells. Lyophilized peptides (Sigma-Merck) were dissolved in DMSO at 10 mM and tested as single peptides (supplementary table 3) or pooled at 1 mM each (supplementary tables 4 and 5). EL4 cells at $10^7$ cell /mL were pulsed with single or pooled peptides at 5 μM each for 2 hours at 37 °C, 5% CO$_2$.

## Flow cytometry

For FC analysis of in vivo samples, organs were smashed into small pieces and then incubated 15 min at 37 °C in agitation with a tissue digestion solution composed of 1 mL IMDM (Corning) supplemented with 0.35 mg/mL collagenase type IV (Sigma-Aldrich/ SCR103), 1 mg/ml dispase II (Gibco/17105041) and 0.2 mg/ml DNAse (Roche/ 11284932001). The tissue was then further dissociated by pipetting and filtered using 40 μm cell strainers (Corning/352340). Single-cell solution was washed with 30 mL of PBS and pelleted for 5 min 300 g 4 °C to remove excess enzymes. Samples were transferred to FACS tubes. Viability of cells was assessed using LIVE/DEAD™ Fixable Blue Dead Cell Stain Kit (Invitrogen) according to manufacturer's recommendation for fixed samples. Upon single cell dissociation, to prevent unspecific staining through binding of the FC receptor, we added to the cells Fc Block (BD Pharmagen). For membrane-bound antigens, samples were stained for 15 minutes on ice. For staining of intracellular proteins, cells were fixed, permeabilized and stained using the True-Nuclear™ Transcription Factor Buffer Set (BioLegend) according to manufacturer's recommendation. For the staining of TCRs specific for the SIINFEKL peptide loaded on MHC class I (H2-Kb), samples were stained with an SIINFEKL-loaded MHC class I tetramer (NIH tetramer core facility) according to manufacturer's instruction. Samples were acquired by using a FACSymphony™ A5 Cell Analyzer (BD Biosciences). For fluorescence activated cell sorting a BD FACSAria Fusion was used. For further information, see supplementary table 6.

For blood samples, after collection 70 μL of blood were moved to a FACS tube and stained with an SIINFEKL-loaded MHC class I tetramer (NIH tetramer core facility) according to manufacturer's instruction. For membrane bound antigens, samples were stained for 15 minutes on ice. Samples were acquired by using a FACSymphony™ A5 Cell Analyzer (BD Biosciences).

We used the following gating strategy to define cell populations by FC. For the lymphoid cell characterization: B cells (CD45$^+$ CD11b$^-$ B220$^+$), CD4 T cells (CD45$^+$ CD11b$^-$ B220$^-$ CD4$^+$), CD8 T cells (CD45$^+$ CD11b$^-$ B220$^-$ CD8$^+$), NK cells (CD45 + , Nkp46 + ), OVA-specific CD8 T cells (CD45$^+$ CD11b$^-$ B220$^-$ CD8$^+$, SIINFEKL$^-$MHC+). Whitin CD8$^+$ T cells or OVA-specific T cells, we defined TEX as EOMES$^+$ PD1$^{high}$ and PEX as TBET$^+$ PD1$^{int}$. For the myeloid cell characterization, we gated myeloid cells as follow: KCs (CD45$^+$ CD11b$^+$ F4/80$^{high}$), granulocytes (CD45$^+$ CD11b$^+$ Ly6g$^{high}$), DC (CD45$^+$CD11b$^+$Ly6g$^-$CD11c$^+$), TAMs (CD45$^+$CD11b$^+$Ly6g$^-$ CD11c$^-$F4/80$^+$), monocytes (CD45$^+$CD11b$^+$Ly6g$^-$ CD11c$^-$ F4/80$^-$).

## Processing of samples for imaging

For IF, tissues collected during necropsy were immediately incubated in a paraformaldehyde solution 4% in PBS (ChemCruz/ 30525-89-4) for 4-12 hours according to tissue size at 4 °C. Afterwards, the PFA was exchanged for a solution of 10% sucrose (Sigma-Aldrich) and 0.02% NaN3 in H$_2$O. After 8-15 h incubation at 4 °C, sucrose solution was increased to 20 %, and to 30% after additional 8-15 h. The organ was then embedded into Killik, O.C.T. Compound embedding medium for cryostat (Bio-Optica/ 05-9801). Sections of 20 nm thickness were prepared and placed on glass slides (Epredia/916155) using a cryostat. And

stored at -20 °C Before staining, sections were dried for 5 minutes at room temperature. For antigen retrieval, slides were incubated for 10 minutes at 95 °C in a preheated water bath in the following solutions: (1) low pH antigen retrieval: 10 mM citric acid in H2O, pH adjusted to pH 6; (2) high pH antigen retrieval: 10 mM Tris base and 1 mM EDTA plus 0.05% tween in H2O, pH adjusted to pH 9. Slides were then cooled down for 15 minutes at room temperature and then washed with PBS 0.1% Triton X-100 (Sigma/9036-19-5) 5 min x 3 times. Blocking was performed using a blocking buffer composed of 5% normal donkey serum, 1% BSA (Sigma-Aldrich/A3912) and 0.1% Triton X-100 (Sigma/9036-19-5) in PBS. After 1 h of blocking at RT, the blocking buffer was replaced by blocking buffer containing primary antibodies at indicated concentrations (see supplementary table 7) and incubated overnight at 4 °C. Sections were then washed with washing buffer (PBS containing 0.1% TritonT X-100) for 5 min × 3 times. Sections were stained with the secondary antibodies in blocking buffer at the indicated concentrations (see supplementary table 7). An incubation for 1 h at room temperature in the dark was performed followed by 3 × 5 min washing steps with washing buffer. For staining of the nuclei, sections were covered with a 1/2000 dilution of Hoechst 33342 solution (Thermo Scientific/62249) in PBS for 2 min. Slides were washed an additional 3 times per 5 min each with washing solution and mounted using Fluoromount-G (SouthernBiotech/0100-01). Images were acquired using an SP8 lightning confocal microscope (Leica Microsystems).

For the histopathologic evaluation of side effects, the indicated organs were collected from mice after euthanasia and fixed in 10% buffered formalin, embedded in paraffin wax, sectioned, and stained with haematoxylin and eosin following OECD Good Laboratory Practices principles, principles of data integrity and applicable GLP SR-TIGET SOPs. Histopathological changes were evaluated by an experienced pathologist and graded on a scale of 1 to 5 as minimal (1), mild (2), moderate (3), marked (4), or severe (5); minimal referred to the least extent discernible and severe the greatest extent possible. The slides were independently reviewed by experienced pathologists, and a consensus was reached on the findings and scores.

## RNA sequencing

RNA was extracted from pelleted cells or tissues using the miRNeasy plus mini Kit (Qiagen/74134). RNA samples from cultured AKTPF, LMs, healthy liver, and intestine of 3 mice were sent to Azenta for library preparation and sequencing with Illumina NovaSeq 2 × 150 bp sequencing.

Pre-processing of the input sequences was done with FastQC (v0.11.6) to assess reads quality and trimmomatic to get rid of low-quality sequences. Then, reads were aligned to the mouse genome assembly (GRCm38) using the STAR software (v2.7.6a) with standard parameters, and abundances were calculated using the Subread featureCounts function (v2.0.1). Differential gene expression analysis was performed using the R/Bioconductor package DESeq2 (v1.30.0), normalizing for library size using DESeq2's median ratios. *P* values were corrected using false discovery rate (FDR), and genes having FDR < 0.05 were considered as differentially expressed.

Variant calling on RNA-Seq data was performed by exploiting alignments to the mouse genome assembly (GRCm38) using the STAR (v2.7.6a). Then, following the GATK "Best Practice Workflows", duplicates were marked using Picard (v2.25.6) MarkDuplicates and GATK (v4.2.0). SplitNCigarReads was used to split reads containing Ns. Variants were then called using HaplotypeCaller (with options --min-base-quality-score 20, --dont-use-soft-clipped-bases, and –standard-min-confidence-threshold-for-calling 20).

Resulting variants were filtered using VariantFiltration based on their 'QualityByDepth' (i.e., -filter 'QD < 2.0'), mapping quality (i.e., -filter 'MQ < 40.0'), genotype quality (i.e., -filter 'GQ < 80.0'), and overall coverage 'DP' (i.e., -filter 'DP < 10'). The final list of variants was then merged with those resulting from the WES experiment.

## WES

Genewiz (Azenta) performed WES using the Agilent Sure Select Mouse All Exon V7 kit and Illumina NovaSeq 2 × 150 bp sequencing, yielding -10 gigabytes (100×) per sample. Data analysis was performed in accordance with GATK "Best Practice Workflows" for variant identification. Initially, FastQC (v0.11.9) was used to assess read quality and trim-galore (v0.6.6) to trim low-quality bases. Alignment to the mouse genome assembly (GRCm38) was done employing BWA (v0.7.17). This was followed by duplicate marking using Picard (v2.25.6) MarkDuplicates. GATK (v4.2.0) BaseRecalibrator + ApplyBQSR was used to recalibrate base quality scores on dbSNP known sites. HaplotypeCaller in GVCF mode was used to call variants in each sample, and variants were merged using CombineGVCFs and genotyped with GenotypeGVCFs. Variant filtering was performed using VariantFiltration based on 'QualityByDepth' (i.e., --filter-expression 'QD < 2.0') and overall coverage 'DP' (i.e., --filter-expression 'DP < 500'). To identify sample-specific private variants, additional filters were applied, removing variants with low genotype quality (i.e., GQ < 80) and low coverage (i.e., DP < 50). The "control" sample served as a germline reference, and its variants were excluded from other samples. The remaining variants were annotated using SnpEff (v5.0) on the canonical isoform from the GRCm38.99 reference database and intersected with those from the RNA-seq experiment.

## Antigen prediction pipeline

WES and RNA variant calling analysis were merged. Only missense mutations identified in at least three out of four datasets (WES and RNA from in vivo and in vitro samples) were considered as input for two different epitope prediction pipelines: antigen garnish and Pvac tools[34,35]. The output of these two pipelines was integrated with the information about the expression of the mutated genes, in transcripts per million (TPM), obtained by the RNA sequencing. The following parameters were then used to select the best-scoring candidates: (i) predicted affinity for MHC-I, (ii) expression in TPM, (iii) agretopicity, and (iv) foreignness score. The top-scoring candidates were then manually confirmed on RNA reads using an integrative genomics viewer.

## Single-cell RNA-seq

Immediately after sacrificing the mice, livers were collected and dissociated into single cells as described above. Single cells were divided into two tubes, one was stained only with CD45, while the second with tetramer, anti-CD3, anti-CD8, anti-CD11b, anti-F4/80, anti-B220 and Hashtag antibodies, then resuspended in MACS buffer containing DAPI. For each mouse, 30,000 CD45+ cells were sorted together with up to 2000 barcoded tetramer⁺ CD8⁺ T cells.

Sorted cells were further processed for scRNA sequencing. ScRNA sequencing was performed using the Next GEM Single Cell 5' GEM Kit v2 from Chromium 10× according to manufacturer's recommendation (User Guide Chromium Next GEM Single Cell 5' Reagent Kits v2). We loaded 20,000 cells belonging to the same sample per reaction. We sequenced 4 samples, 150 bp paired-end reads in a NovaSeq 6000 Illumina apparatus. Base call files obtained as a result of the Illumina sequencing were converted into FASTQ files, aligned to the mouse reference genome *(GRCm38)*, and quantified with the 10x Genomics Cell Ranger Software (v7.2.0) using default parameters. The resulting data was imported into R and processed with the Seurat package (v5.0.1). All samples were merged into a single object and processed to remove cells with a low sequencing quality, those with a feature count below 600 and above 9000, as well as cells with a fraction of mitochondrial genes higher than 10 %. Samples were demultiplexed using the hashtag information using the Seurat function HTODemux, and cells classified as HTO doublets were filtered out. At the same time, the scDblFinder (v1.16.0) package was employed to annotate doublets of cells and exclude them from the following analyses. UMI-counts were

then normalized and scaled using the Seurat functions NormalizeData (normalization method "LogNormalize") and ScaleData. During this scaling step, unwanted sources of variation were regressed out. These sources included the number of detected transcripts per cell, the percentage of transcripts originating from mitochondria, and the difference between the scores of the cell cycle phases S and G2/M calculated for each cell. A principal component (PC) analysis with 100 PCs was performed for dimensional reduction, and afterwards, we computed a UMAP representation using the top 35 PCs. Additionally, cell clusters were identified based on these top 35 PCs using a resolution parameter of either 0.6 or 1.2. Marker genes for each cluster were obtained using the FindAllMarkers Seurat function, and finally, clusters were manually annotated and curated. Following GSEA analysis and visualization were performed on R using packages fgsea and enrichplot, respectively. For GSEA the gene sets from https://www.gsea-msigdb.org/gsea/msigdb/genesets.jsp were used. The CD8+ T cell exhaustion signature, termed Exhaustion and the IFNα signature were retrieved from publications[70,71]. TCR sequencing was performed using Next GEM Single Cell 5' GEM Kit v2 in combination with the Single Cell Mouse TCR Amplification Kit. The downstream analysis was performed as described above. TCR analysis was done with the R/Bioconductor package scRepertoire (v1.12.0) to identify clonotypes in each sample and compute their frequency (considering the amino acid sequence of the CDR3 region of the beta chain). TCR clonotypes were classified as unique (1 cell containing a specific TCR clone), small (2–5 cells containing the same TCR), large (6 to 30 cells containing the same TCR), and hyperexpanded (>30 cells containing the same TCR), according to their numerosity.

In order to select genes to represent in dot plots, we perform GSEA comparing treated groups vs control to identify biological pathways and immune functions modulated by our treatment. Building on the results, we then manually selected well-known genes involved in these biological functions. For example, in Fig. 2E we are plotting the expression level (i.e., fraction of cells and scaled average expression) of selected genes expressed by all APCs in all the distinct groups. P values were calculated by comparing the expression between each treatment group and the IiOVA group.

## Cell-to-cell interaction analysis

To explore interactions among TAMs, DCs, and CD4+ and CD8+ T cell, we employed MultiNicheNet, a computational method for differential cell-to-cell communication analysis[37]. As a first step, we extracted abundance and expression information from sender and receiver cell types, combining this expression of ligands in the senders with the corresponding receptors in the receivers. A set of affected target genes in the receiver was defined based on genome-wide differential expression analysis of receiver and sender cell types and subsequently used to predict the MultiNicheNet ligand activities and MultiNicheNet ligand-target links. MultiNicheNet ligand-target links were finally used to prioritize all sender/ligand–receiver/receptor pairs and calculate their expression correlation with the predicted target genes.

## MERSCOPE tissue collection, sectioning, and quality controls

Liver samples containing tumors were snap frozen, preserved in optimal cutting temperature compound, and kept at −80 °C until sectioning. Samples were sectioned at −20 °C on a cryostat, and 10-µm thick sections were placed on a MERSCOPE slide. The sectioning procedures were performed in an RNAse-free environment, following manufactures instructions. Ten µm sections were additionally collected into Eppendorf tubes to extract total RNA (QIAGEN RNA extraction KIT) and verify RNA quality by a microcapillary system (TAPE station, Agilent). All the tissue sections processed showed total RNA with a DV200 value, representing the relative abundance of fragments with a size greater than 200 nt, equal to 50% or more.

## MERSCOPE probe panel desing and sample acquisition

To profile spatial gene expression in liver sample, we selected 500 genes, accordingly to data previously generated by bulk or single-cell RNA sequencing. The list of genes was filtered by using the Gene Panel Design Portal (https://portal.vizgen.com) to ensure that each gene was sufficiently long to allow enough encoding probes to bind, and that the entire gene panel meets the abundance threshold to avoid optical crowding for MERSCOPE imaging. This resulted in a final panel of 500 genes and 50 blank barcodes.

Tissue sections on MERSCOPE slide were processed according to manufactures' instruction (MERSCOPE User Guide, Fresh and Fixed Frozen Tissue sample preparation REV D, Vizgen) with provided buffers and reagents, and loaded for imaging on the MERSCOPE instrument.

## Distance calculation of transcripts in MERFISH data

To investigate the spatial relationships among genes in our MERFISH datasets, a custom R script was developed to quantify the distance between individual transcripts. Specifically, a 500 × 500 matrix was generated to record the distance score between each pair of the 500 genes present in our MERFISH datasets.

The script iteratively analyzed each gene pair, counting the number of "close" transcripts between them using Euclidean distance as a metric. Two transcripts were considered "close" if their Euclidean distance was below the distance threshold of 30 µm. This threshold distance was chosen as the radius of 30 µm generates a circular area that contains a small number of cells, typically 5–10 cells, allowing for the investigation of spatial relationships at the cellular level. To maintain biological relevance, only transcripts originating from different cells were considered, excluding those sharing the same cell ID.

Importantly, to accurately estimate the fraction of close transcripts, if a transcript of gene A was close to multiple transcripts of gene B it was counted as a single close interaction and, for this reason, the resulting matrix is asymmetrical. To obtain a normalized distance score ranging from 0 to 1, the count of close transcripts was divided by the combined number of transcripts from both genes and then multiplied by two.

We selected a subset of genes that identify specific cell types, and we visualized their normalized distance score using a heatmap. To compare the results of the TA33 and TA33.Combo groups, we subtracted the TA33.Combo normalized matrix from the TA33 one. Therefore, positive values in the resulting matrix indicate closer proximity of the two genes in the TA33.Combo compared to the TA33 condition.

## Survival analysis

Survival data and gene expression data for various cancer types were obtained from The Cancer Genome Atlas (TCGA) database. For each cancer type, patients were filtered based on their tumor type. The expression levels of the IIT signature genes were extracted and log2-normalized. The mean expression score for each patient was calculated by summing the log2-transformed expression values across the selected genes. Patients were stratified into high and low signature score groups based on the 50th percentile of their gene expression scores. Specifically, patients with a gene expression score above the 50th percentile were classified as the high signature group, while the remaining patients were classified as the low signature group. Survival curves were generated for the high and low signature groups using the Kaplan-Meier method. The log-rank test was used to compare survival differences between the two groups. Additionally, a Cox proportional hazards model was fitted to evaluate the association between the signature score and overall survival.

To assess the expression of the IIT signature in melanoma patients treated with a-PD1 we exploited a previously published dataset[72].

## Statistics and reproducibility

Based on previous studies, groups of 5–10 mice were used to detect statistical differences between groups. The sample size for each experiment is indicated in the figure legends. Samples were assigned unique identifiers for blind analysis. Randomization methods were not employed, as the groups were homogeneous and composed of equivalent animals in terms of age, sex, and genotype. All mice used in the study were included, except those with unrelated health issues. No outliers were excluded. Comparisons between two independent groups were performed with the Mann–Whitney test. Comparisons among more than two groups were performed using the Kruskal–Wallis test with post-hoc analysis through Dunn's test for multiple comparisons. For some experiments, comparisons were performed versus the control group, as indicated in the figure legend. In all the other experiments, all possible comparisons were made, and statistics were indicated in the figures only when $p$ values were lower than 0.05. Groups with fewer than five mice were excluded from the statistical analysis. The correlation between the two variables was performed with Spearman's correlation coefficient. Statistical descriptors are always indicated in the figure legends. In VCN and OVA gene expression analyses of the tumor, mice that completely eradicated the tumor were reported with a value of 0. Values below the dark gray area in OVA-specific CD8$^+$ T cell plots could not be distinguished from background noise and were excluded from further analyses. All experiments have been performed once and are shown in the manuscript.

In all analysis, the significant level was set at 0.05. All statistical analyses were performed with GraphPad prism or using R version 4.1.2 [http://www.R-project. org/].

Figures were created using Prism 9 Version 10.1.0.

FC analyses were performed using FlowJo version 10.8.1.

Illustrations in Figs. 1b, g, 5f, h, and 7d, and supplementary Figs. 7b, 9b, and 10a were Created in BioRender. Squadrito, M. (2025) https://BioRender.com/z11e077 and edited in Adobe Illustrator.

## Reporting summary

Further information on research design is available in the Nature Portfolio Reporting Summary linked to this article.

## Data availability

The MERFISH, single-cell RNA sequencing and bulk RNA sequencing data have been deposited in the GEO repository under the accession number GSE273615. Additionally, the WES data have been uploaded to the ENA portal with the accession number PRJEB78386. Source data are provided with this paper.

## Code availability

Code is available at the following link: http://www.bioinfotiget.it/gitlab/custom/notaro_mouse_lm_2025.

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

## Acknowledgements

This work was supported by grants from the Lombardy Foundation for Biomedical Research (FRRB) "Early Career Award" grant (GenTooControl, ID 1751658), the Cariplo Foundation Grant (2019-1834), the Italian Fund for Science (FIS00002240) and the Italian Association for Cancer Research (AIRC) grant ("5 per Mille", 22737, "Italy Post-Doc", 29619, IG, 30759). We thank the NIH Tetramer Core Facility for providing APC-conjugated H2-Kb-SIINFEKL tetramer for OVA-specific T-cell detection. We thank the IRCCS San Raffaele Hospital Flow Cytometry Facility (FRACTAL), the IRCCS San Raffaele Center for Omics Sciences (COSR), the IRCCS San Raffaele Hospital Advanced Light and Electron Microscopy BioImaging Center (ALEMBIC), the San Raffaele Telethon Institute for Gene Therapy (SR-TIGET) Process Development Laboratory (PDL) for production of medium scale purified LV stocks.

## Author contributions

M.N. performed research, interpreted data, and wrote the manuscript. M.B., C.B., G.G., T.K., and C.M.M. contributed to the research and interpreted data. M.N., S.B., M.M., and I.M. performed bioinformatic analyses. S.I., A.A., and S.G. performed ELISpot assays. T.C. performed MRI scans. M.G. and R.O. contributed to single-cell RNA sequencing and MERFISH sample preparation, processing, and analysis. P.C., F.S., S.D.I., performed histopathology analyses. P.M.V.R. contributed to statistical analyses. L.N. interpreted data, supervised research, and wrote the manuscript. M.L.S. designed the study, interpreted data, supervised research, wrote the manuscript, and coordinated the work.

## Competing interests

M.L.S., L.N., M.N., and T.K. are inventors on a patent on KC-directed gene transfer, and L.N. is an inventor on patents on miRNA-regulated LV technology filed and managed by the San Raffaele Scientific Institute and the Telethon Foundation. The remaining authors declare no competing interests.
