## [Transparent Peer Review file · Nature Communications]

In vivo armed macrophages curb liver metastasis through tumor reactive T cell rejuvenation

Corresponding Author: Dr Mario Leonardo Squadrito

Version 0:

Reviewer comments:

Reviewer #1

(Remarks to the Author)

In this manuscript, Notaro et al. used a lentiviral vector (LV) platform for delivery of tumor antigen (TA), IFN α , and IL-12 for the treatment of liver metastasis, which remains a significant clinical issue with low response to immunotherapy. The authors showed that the combination of TA with IFN α and IL-12 resulted in superior therapeutic effects on their experimental mouse models of liver metastasis. The study design is simple but the authors revealed in depth that the mechanisms involve activation of macrophages, instead of T helper cells, in the liver and rewiring of the tumor microenvironment, leading to rejuvenated tumor-reactive T cells as evidenced by a lot of single-cell sequencing data. Overall, this manuscript provided a potent gene transfer therapeutic strategy and insights into the anti-tumor mechanism. Below are several major issues that need to be addressed before publication of this manuscript.

Major:

1. The authors stated that their LV is liver macrophage-selective, which is the basis for activating macrophages in the liver and modulating the liver metastasis microenvironment. However, there is little demonstration of the biodistribution either on the organ levels or the tissue-cellular levels, particularly for the liver macrophages. It may be possible that the transfection to other cells, such as hepatocytes and LSECs, produced TAs and cytokines that activated the macrophage in the liver. DCs were also activated in the liver as well (Figure 2). It may also be important that the LV transfection in tumor cells induces direct anti-tumor effect. Moreover, the biodistribution study could be also associated with the biosafety issue as the authors only tested the inflammatory cytokine levels in blood and the transaminases in the liver but not other assays related to safety of other tissues.
2. The authors compared low dose (107 TU/mouse) and high dose (108 TU/mouse) of TA and found that the high dose showed no therapeutic effects, although the OVA-specific CD8 T cells were significantly elevated in the liver. The authors explain that high and sustained levels of TA expression (which is not demonstrated in the manuscript) may lead to defective CD8 T cell activation as evidenced by a high fraction of TEX CD8 T cells. More supporting data may be needed to further clarify this. For example, the TA expression at low vs. high dose of LV needs to be examined. The authors only tested two different doses and the dose of 107TU/mouse may not be the optimal one. In addition, the authors measured s.c. tumor growth but not the treatment effects for the live metastasis when comparing the two doses.
3. For the combo treatment, did the authors load LVs with all three plasmids, or mixed separate LVs loaded with different plasmids? If the latter, what is the ratio and what is the dose for a single treatment group? For example, the cytokine level of IL-12 is lower for the OVA+IFN α +IL12 combo group than the OVA+IL12 group (Fig S1U).
4. The authors stated that the addition of TA to IFN α + IL-12 is necessary for the superior tumor inhibition effects. However, current data may not be sufficient to support this claim. It seems more likely that the addition of IFN α + IL-12 enhanced the generation of TA-specific CD8 T cells. First, the low dose OVA group showed a minimal increase of the OVA-specific CD8 T cells as compared to the control group $P = 0.0476$ (Figure 1E), indicating OVA alone can only slightly induce TA-specific CD8 T cells. Second, for the combo cytokine treatment without tumor antigen (IFN α + IL-12), the therapeutic effect is also superior (Figure 5A), and there is no difference for all aspects in terms of cytokine levels, T cell activation and, more importantly, tumor volume (Figure 5A, S5). The level of TEX T cells is even lower in the combo group (Figure 5C). These results demonstrate that the two cytokines may be the key to induce tumor-reactive T cells, although the authors found no antigen-specific CD8 T cells (Figure 5E). It seems that TA (e.g., Trp2)-specific CD8T cells had little contribution to the anti-tumor efficacy.

Minor:

5. Flow gating strategy needs to be provided
6. The variability of tumor growth for the control group in Figure 7E is large. A larger n number may be needed.
7. In some cases, 'α' was written as 'a' in the manuscript.

Reviewer #2

(Remarks to the Author)

This study presented a new method for gene therapy for the treatment of liver metastasis. The technique employs a lentiviral vector (LV) to ensure the coordinated expression of tumor antigens (TAs), IFN α , and IL-12 in the liver and tumor-associated macrophages (TAMs) through a single intravenous administration. The LVs incorporate a mannose receptor C type-1 (Mrc1) promoter, which is expressed mainly in resident macrophages, and microRNA target sequences that prevent transgene off-target expression in other cell types in the liver. The activated liver macrophages that facilitate the concurrent expression of TA and cytokines have proven to be effective in inhibiting tumor growth in different cancer models that form liver metastases. The majority of the mice showed complete response to the therapy. The inhibition of the tumors is linked to a wide-ranging reprogramming of the tumor microenvironment (TME) as well as the activation of TA reactive CD8 $^+$ T cells. In addition to presenting a promising therapeutic tool, this research provides valuable insights into the mechanisms that promote immune activation and reprogramming in the metastatic liver cancer.

Major concerns:

1. The methodology of lentiviral vector (LV) delivery could be more clearly described in the study. It would be helpful to provide a detailed description of the LV administration process, including the dose, timing, and route of administration. This information would allow other researchers to replicate the study and also help in the development of future studies.
2. The study mainly used an artificial metastasis model by implanting colorectal and melanoma cells in the liver of mice. A natural metastasis model could have been used to better reflect the process of liver metastasis. The use of a more natural model would provide a more accurate representation of the effectiveness of the therapy in treating liver metastasis. This current model only demonstrates the effects of the LV therapies after metastatic seeding at the soil but not the entire process.
3. The study used a highly immunogenic system (OVA) for the analysis of tumor antigen (TA) expression. To make the study more applicable to natural metastasis models, the use of more natural antigens, such as the PMEL-T cell system, should be considered. This would provide a more accurate representation of the effectiveness of the therapy in treating liver metastasis.
4. The study lacks a comprehensive analysis of the macrophages in the entire study. While the authors focused on the T cells, macrophages are the target cells that elicit the anti-tumor response. A more comprehensive analysis of the changes in macrophages under the influence of the LV system could be validated by scRNA sequencing followed by flow cytometry studies. This would provide a better understanding of the therapy's impact on the macrophages' properties and how they contribute to the anti-tumor response. Interaction strength between different cell types can also be evaluated.

Reviewer #3

(Remarks to the Author)

In the manuscript "In vivo armed macrophages curb liver metastasis through tumor reactive T cell rejuvenation" from Notaro et al, the authors used mouse models combined with single cell rna sequencing to obtain important insights on the role of the immune system in liver metastasis. My comments are aimed specifically at the single cell rna sequencing parts of the manuscript:

- 1) "To better investigate the effect of simultaneous IFN and IL-12 delivery on immune cells, we performed single cell RNA sequencing (scRNA-seq) on CD45 $^+$ cells isolated from healthy liver parenchyma and matched LMs of treated mice."
 - Can you elaborate on the relatively high number (at least visually) of non CD45 $^+$ cells identified? Can you display somewhere in the text or supplementary which % they indicate?
- 2) "All cells were manually annotated based on their transcriptomic profile"
 - Can you include plots of known markers for all clusters? This makes the reading a bit easier instead of checking the supplementary tables.
- 3) "Analyzing together all APC cell types, we observed in both liver and LM that OVA.Ifn α increased the expression of genes involved in MHC-I antigen presentation, such as MHC-I subunits (H2-Q7, H2-T22, H2-Q4, H2-K1, H2-M3, H2-D1) and peptide transporters (Tap1, Tap2), and of interferon 208 stimulated genes (Irf44, Irf7, Isg20, Oas1a, Oas1g) compared to the liOVA group (Fig. 2, E and 209 F)."
 - For sake of clarity, were these genes identified exactly how? By comparing different clusters to each other? Or all cells from specific group versus each other? Can you elaborate on how they were identified?
 - Please harmonize cell type nomenclature. In fig 2G you refer to granulocytes, which I assume are the neutrophils in 2C and 2D. If DCs were combined, please clarify as well.
- 4) "expanded TCR clonotypes were mostly shared between the liver and LM. Of note, CD8 $^+$ T cells bearing expanded and shared TCR clonotypes clustered with CD8 $^+$ T effector cells in the UMAP plot, representing putative treatment-induced tumor specific CD8 $^+$ T cells (Fig. 6B)"
 - The authors show an interesting enrichment of enriched clonotypes in very well defined cell clusters. However, as shown by Sundell et al 2022 DOI: 10.1093/bfpg/elac044, sometimes TCR related genes are captured by both 3' and 5' single cell maseq protocols. These genes can eventually lead to biased cluster identification, which in the context of intense clonotype expansion can be particularly harmful. I would recommend the authors reanalyse the data after exclusion of TCR related

genes.

5) "Base call files obtained as result from the Illumina sequencing were converted into FASTQ files, aligned to the mouse reference genome, and quantified with the 10x Genomics Cell Ranger Software (v7.2.0) using default parameters."
- Which genome version was used?

Reviewer #4

(Remarks to the Author)

Version 1:

Reviewer comments:

Reviewer #1

(Remarks to the Author)

The authors have adequately addressed my comments. This manuscript can be accepted for publication.

Reviewer #2

(Remarks to the Author)

All questions addressed. Thank you for the revision.

Reviewer #4

(Remarks to the Author)

Reviewer #5

(Remarks to the Author)

The authors sufficiently addressed my concerns

Reviewer #6

(Remarks to the Author)

The authors have addressed the comments raised by the previous Reviewer #3 regarding bioinformatic analysis. I have only one minor comment: The co-localization score heatmaps (supplementary Fig. 14 C and D) are somewhat unclear due to the absence of row and column labels, as well as a legend title. Adding these elements would improve readability.

Point-by-point response to Reviewers

We thank the Editor and the Reviewers for the positive feedback and insightful review of our manuscript. In response to the constructive advice received, we revised the manuscript including additional *in vitro* and *in vivo* experiments, new bioinformatic analysis, text and figures editing to improve clarity as well as an expanded material and methods section to provide clearer methodology description. Overall, we think that these changes have largely improved the manuscript.

The new results, which we have obtained during the revision of the manuscript are indicated below.

1. We performed a dose-response study *in vivo* to investigate the anti-tumoral effects of distinct doses of the liver targeted liOVA LV in a mouse model of colorectal cancer liver metastases (LM). To this aim, we compared the effects of a low dose (i.e., 10^6 TU/mouse), an intermediate dose (mid, i.e., 10^7 TU/mouse, previously referred to as a low dose), or a high dose (i.e., 10^8 TU/mouse) of liOVA LV before challenging the mice with MC38.OVA cells intrahepatically, to generate experimental LM. These doses were selected strategically to span a biologically relevant range while considering key technical and biological factors. The low dose of 10^6 TU/mouse represents the minimal effective dose, as doses below this threshold fail to achieve sufficient transduction likely due to complement inhibition of the LV, as previously observed ¹. The intermediate dose of 10^7 TU/mouse was included to investigate potential dose-dependent effects on anti-tumoral activity. The high dose of 10^8 TU/mouse was selected based on feasibility for translational applications and its prior demonstration as a well-tolerated dose in preclinical settings ¹. Interestingly, only treatment with intermediate (mid) dose of liOVA LV resulted in a significant reduction in tumor growth, with 5/9 mice completely eradicating LM (**Fig. 1C**). These new results confirm our previous findings, supporting the employment of the mid dose (previously referred as low dose) for achieving tumor specific T cell activation.
2. We performed a prophylactic experiment, which, at least in part, recapitulates the clinical scenario where primary tumors are removed from CRC patients, and circulating cancer cells or micrometastases give rise to metastatic recurrences. To this aim, we treated mice with TA33 in combination with IL-12 and IFN α (TA33.Combo LV) before tumor challenge. To model spontaneous metastatic seeding, we injected CRC cancer cells in the spleen of the mice 7 days upon LV treatment and analysed mouse survival. In addition to this, to assess long term effects of the treatment and potential biosafety concerns, we tracked mice health conditions, such as weight and blood parameters by hemocytometric analysis. At an established endpoint at 4 months after LV treatment, we performed a comprehensive histological analysis to evaluate any potential tissue alterations or adverse effects induced by the LV treatment and consequent cytokine expression. Histopathological evaluation of the liver confirmed absence of LM in all TA33.Combo LV-treated mice. Importantly, no significant treatment-related alterations were observed in lung, bone marrow and mesenteric lymph nodes, confirming the safety and tolerability of this combination treatment.
3. We performed an experiment where we splenectomized the mice before tumor placement and LV treatment. In these mice, OVA.Combo treatment favored tumor control and development of OVA specific CD8+ T cells in circulation and the liver even after splenectomy (**Supplementary fig.3, H to J**). These new data suggest that the

therapeutic effects of the concurrent delivery of TA and cytokines can be independent of splenic immune cell populations.

4. To better investigate the impact of our LV treatment on tumor macrophages we expanded our single cell RNA sequencing analysis to better characterize macrophage subsets (**Supplementary fig. 5C**). We found that simultaneous co-delivery of cytokines and TA did not significantly alter the relative abundance of macrophage subsets in the tumor microenvironment (TME) as compared to delivery of TA only (**Supplementary fig. 5D**). However, OVA.Combo treatment increased the expression of genes associated with antigen presentation (both MHC-I and MHC-II) and cytokine stimulation, while reducing the expression of pro-tumoral genes in most of the subsets we defined (**Supplementary fig. 5E**).
5. We analyzed the impact of our LV treatment on tumor myeloid cells, including tumor-associated macrophages, upon OVA.Combo treatment through multicolor flow cytometry, confirming enhanced expression of MHC-II molecules, co-stimulatory receptors as well as other markers of immune activation (**Supplementary fig. 5, F to K**).
6. We excluded TCR genes from the clustering method used in our single cell RNA sequencing analysis by removing 254 TCR genes as previously described ². We thoroughly compared the results obtained with and without TCR genes and found that the removal of TCR genes did not significantly impact the analysis, including the unbiased cell clustering and identification (**Supplementary table R1**).
7. To assess the correlation between LV dose and the extent of TA presentation in MHC class I (MHC-I) in macrophages, we transduced an immortalized cell line of Kupffer cells with increasing doses of li.OVA LV and monitored by flow cytometry the presentation of the OVA peptide SIINFEKL on the Kupffer cell's MHC-I complex. We observed a dose dependent increase in the levels of SIINFEKL-MHC-I⁺ cells as well as an increase in the signal in term of MFI.

As agreed with the Editor, we did not conduct additional experiments using new genetic models of LM beyond the three mouse models already presented in the manuscript. While we acknowledge that employing genetic models that spontaneously develop LM could provide valuable insights to further support the translational relevance of our findings and expand the mechanistic characterization of our treatment. Unfortunately, genetic models of LM are challenging to implement; they often exhibit high inherent variability, and only a small fraction of mice typically develop tumors. Additionally, such models or models where transplantable primary tumors give rise to spontaneous metastases would require the removal of the primary tumor, identification of novel tumor antigens, and time for liver tumor masses to grow, all of which present significant challenges and require surgical procedures that at the moment we are not able to perform. Moreover, we believe that the results obtained with our current mouse models are sufficient to robustly support our conclusions.

We have also revised the manuscript Methods section and the Figure legends to enhance clarity and include the new results. Edited sections in the manuscript are in dark red.

We have included a new author, Sara Degl'Innocenti, who performed histopathological analysis on mouse samples.

Below, we provide a detailed, point-by-point response to address each specific comment. For clarity, Reviewer's comments are dark blue, while our responses are presented in black. Figures and tables that were generated to address Reviewers' comments, but that were not incorporated in the manuscript are indicated as **Fig.R** or **Table R**. Please note, we are providing tables as a single excel file to enable easier data analysis if necessary.

Reviewer's comment: **dark blue**

Our response: **black**

Reviewer #1:

In this manuscript, Notaro et al. used a lentiviral vector (LV) platform for delivery of tumor antigen (TA), IFN α , and IL-12 for the treatment of liver metastasis, which remains a significant clinical issue with low response to immunotherapy. The authors showed that the combination of TA with IFN α and IL-12 resulted in superior therapeutic effects on their experimental mouse models of liver metastasis. The study design is simple but the authors revealed in depth that the mechanisms involve activation of macrophages, instead of T helper cells, in the liver and rewiring of the tumor microenvironment, leading to rejuvenated tumor-reactive T cells as evidenced by a lot of single-cell sequencing data. Overall, this manuscript provided a potent gene transfer therapeutic strategy and insights into the anti-tumor mechanism. Below are several major issues that need to be addressed before publication of this manuscript.

Major:

1. The authors stated that their LV is liver macrophage-selective, which is the basis for activating macrophages in the liver and modulating the liver metastasis microenvironment. However, there is little demonstration of the biodistribution either on the organ levels or the tissue-cellular levels, particularly for the liver macrophages. It may be possible that the transfection to other cells, such as hepatocytes and LSECs, produced TAs and cytokines that activated the macrophage in the liver. DCs were also activated in the liver as well (Figure 2). It may also be important that the LV transfection in tumor cells induces direct anti-tumor effect. Moreover, the biodistribution study could be also associated with the biosafety issue as the authors only tested the inflammatory cytokine levels in blood and the transaminases in the liver but not other assays related to safety of other tissues.

We thank the Reviewer for raising this important point. We apologize for not clearly referencing our prior work investigating the biodistribution of the LV platform ¹. In our previous work, we extensively characterized the biodistribution of LVs upon systemic delivery to immunocompetent mice as well as the expression pattern of a transgene under the transcriptional control of the mannose receptor c-type 1 (*Mrc1*) promoter and post-transcriptional control of microRNA (miRNA) target sites for miR-122-5p and miR-126-3p. Previously, we showed that the transgene expression obtained by employing this LV-based platform is selective for liver macrophages, with negligible expression in splenic MRC1+ macrophages. Of note, we showed no expression in other biological compartments such as blood, bone marrow, lung, sub-iliac lymph nodes, small intestine, or brain. In our previous work, we also analyzed transgene expression in livers bearing colorectal cancer (CRC) metastases and found that cancer cells were virtually untransduced. This information is now more clearly referenced in the revised manuscript.

Regarding the potential biosafety concerns, we agree with the Reviewer that measuring cytokine levels in the blood and transaminases alone may not provide a complete

assessment of the safety profile of the LV treatment. To address this, we tracked mice health conditions, such as weight and blood parameters by hemocytometric analysis after TA33.Combo treatment. At an established endpoint at 4 months after LV treatment, we performed a comprehensive histological analysis to evaluate any potential tissue alterations or adverse effects induced by the LV treatment and consequent cytokine expression. These analyses expand and confirm our previous findings, highlighting normal weight gain over a period of almost 4 months and normal levels of transaminases 7-week post treatment (**Supplementary fig. 10, D and E**). Hemocytometric analysis revealed a mild decrease in red blood cell (RBC), white blood cell (WBC) and lymphocyte counts in mice treated with TA33.Combo compared to control, which was associated with an increase in the reticulocytes counts, in the first 2 months post treatment (**Supplementary fig. 10, F and G**). These parameters returned to levels comparable to tumor-free control mice toward the end of the experiment concomitantly with a decrease in cytokine levels observed in the plasma and clearance of transduced cells from the liver (**Supplementary fig. 10, H and I**). Histopathological evaluation of the liver confirmed absence of liver metastases (LM) and highlighted a slight increase in mixed cell inflammatory infiltrate and extramedullary hematopoiesis compared to tumor-free control mice (**Supplementary fig. 10J**). These minimal changes are most likely secondary to the immune stimulation caused by the treatment. Of note we observed minimal to moderate inflammation in the splenic capsule and minimal to moderate lymphoid hyperplasia in few mice in the LV-treated group. These findings may be at least in part associated to the surgical procedure that was employed to challenge the mice with experimental LM through intrasplenic injection. Importantly, no significant alterations were observed in lung, bone marrow and mesenteric lymph nodes, confirming the safety and tolerability of this combination treatment.

Of note, IL-12 delivery has been associated with liver and systemic toxicity³. However, most studies addressing IL-12 toxicology have focused on the delivery of the recombinant protein. Our LV platform enables sustained and more stable cytokine levels compared to the recombinant protein administration, offering an alternative strategy which limits toxic side effects. We think that our LV-based approach could potentially reduce IL-12-associated toxicity, even in combination with other cytokines, such as IFN α , or immunotherapeutic interventions. Nonetheless, it is important to acknowledge that higher doses may still result in toxic effects similar to those observed with recombinant cytokine delivery, as reported in previous studies.

2. The authors compared low dose (10^7 TU/mouse) and high dose (10^8 TU/mouse) of TA and found that the high dose showed no therapeutic effects, although the OVA-specific CD8 T cells were significantly elevated in the liver. The authors explain that high and sustained levels of TA expression (which is not demonstrated in the manuscript) may lead to defective CD8 T cell activation as evidenced by a high fraction of TEX CD8 T cells. More supporting data may be needed to further clarify this. For example, the TA expression at low vs. high dose of LV needs to be examined. The authors only tested two different doses and the dose of 10^7 TU/mouse may not be the optimal one. In addition, the authors measured s.c. tumor growth but not the treatment effects for the live metastasis when comparing the two doses.

We thank the Reviewer for these valuable comments. We appreciate the suggestion to provide additional supporting data to clarify the impact of distinct LV doses on antigen expression and CD8⁺ T cell activation. To address this comment, we first investigated the correlation between the LV dose and the extent of TA presentation in MHC class I (MHC-I) in macrophages. We transduced an immortalized cell line of Kupffer cells with increasing doses of li.OVA LV and monitored by flow cytometry the presentation of the OVA peptide SIINFEKL on the Kupffer cell's MHC-I complex. We observed a dose-dependent increase in the levels of SIINFEKL-

MHC-I⁺ cells as well as an increase in the signal intensity in terms of median fluorescence intensity (MFI) (Figure for the Reviewer, **Fig. R1A**). This result suggests that higher doses of LV, besides reaching a higher number of cells, might lead to presentation of the TA-derived peptide on a higher fraction of MHC complexes per cell.

Fig.R1. An immortalized KC cell line was transduced with different doses of liOVA LV. After 7 days transduced cells were harvested and stained with SIINFEKL-MHC-I for flow cytometric analysis ($n=3$ biological replicates).

We then performed a dose-response study *in vivo* to investigate the anti-tumoral effects of distinct doses of liOVA LV in a mouse model of LM. To this aim, we compared the effects of a low dose (i.e., 10^6 TU/mouse), an intermediate dose (mid, i.e., 10^7 TU/mouse, previously referred to as a low dose), or a high dose (i.e., 10^8 TU/mouse) of liOVA LV before challenging the mice with MC38.OVA intrahepatically to generate experimental LM. Interestingly, only treatment with intermediate (mid) dose of liOVA LV resulted in a significant reduction in tumor growth, with 5/9 mice completely eradicating LM (**Fig. 1C**). We observed expansion of OVA specific CD8⁺ T cells in the blood and liver of liOVA LV-treated mice in a dose-dependent fashion (**Fig. 1, D and E**). Moreover, liOVA delivery induced a dose-dependent accumulation of terminally exhausted (TEX) OVA specific CD8⁺ T cells, accompanied by a corresponding reduction in progenitor exhausted (PEX) OVA specific cells (**Fig. 1F**). These new results further underscore the importance of achieving an optimal antigen load to effectively activate tumor-specific CD8⁺ T cells in the liver and maintain a functional balance between expansion of TEX and PEX subsets. These new results confirm our previous findings, supporting the employment of an intermediate dose (previously referred as low dose) of TA for achieving tumor specific T cell activation.

We also agree with the Reviewer that it is counterintuitive that higher dose of TA may lead to a less robust anti-tumoral effect. Indeed, it has been observed that increased doses of TA presentation may be beneficial in some experimental settings^{4,5}. However, it also has been observed that sustained and robust expression of antigens can lead to defective T cell activation, hence reduced tumor control^{6,7,8}. It is possible that in our setting of LV-based liver-directed delivery of TA, both the extent of cells expressing the antigen, the level of antigen presentation per cell, the immunogenicity of the tumor antigen and the immune features of the liver may contribute to the observed behavior. Of note, albeit understating the nuances of antigen presentation may be of key importance to design better liver-directed vaccines, understanding these mechanisms falls beyond the scope of our study.

3. For the combo treatment, did the authors load LVs with all three plasmids, or mixed separate LVs loaded with different plasmids? If the latter, what is the ratio and what is the dose for a

single treatment group? For example, the cytokine level of IL-12 is lower for the OVA+IFN α +IL12 combo group than the OVA+IL12 group (Fig S1U).

We apologize for not clearly explaining this point, which was initially indicated in the Materials and Methods section but that not indicated in the result section. We have now included this information also in the Results section.

To clarify, mice were treated with a mix of three different LVs. In brief, LVs were produced in HEK293T, which were transfected with distinct transfer plasmid (either carrying TA, IFN α or IL-12) along with third-generation packaging plasmids, pGag-Pol, pRSV-REV, pAdVantage as well as an envelope protein plasmid encoding VSV.G. After 14 h from the transfection, media was replaced, and after additional 30 h, the LV-containing supernatant was harvested, filtered and ultracentrifuged. The pellet containing the LV particles was then resuspended in fresh PBS and stored at -80°C. The produced LVs were then titrated on HEK293T cells to determine the TU/mL of the produced vector. For the TA.Combo treatment, three distinct LV preparation (*i.e* TA, IFN α , IL-12) were mixed to achieve the desired dose (10⁷ TU/mouse for TA, 10⁸ TU/mouse for IFN α , 10⁷ TU/mouse for IL-12) in a final volume of 200 μ L, which was then injected intravenously in mice.

We administered 10⁷ TU/mouse for IL-12 and 10⁸ TU/mouse for IFN α in all experiments involving these lentiviral vectors (LVs). When IL-12 and IFN α were used in combination, we maintained the same doses (*i.e.*, 10⁷ TU/mouse for IL-12 and 10⁸ TU/mouse for IFN α). As the Reviewer correctly noted, the cytokine output observed for IFN α and IL-12 differed slightly when the LVs encoding these cytokines were used individually versus in combination (**Fig. S3B**).

This minor variation may be attributed to:

1. A positive feedback loop between IL-12 and IFN α , potentially amplifying IFN α levels, as seen in the OVA.Combo group compared to OVA.Ifna.
2. Competition between LVs during transduction, which could reduce the overall transduction efficiency of individual components in the combination treatment.
3. Additional, less-characterized factors triggered by either IL-12 or IFN α that might influence the stability or clearance of these cytokines in circulation.

These potential mechanisms warrant further investigation to fully understand the interplay between these cytokines when delivered via LVs.

4. The authors stated that the addition of TA to IFN α + IL-12 is necessary for the superior tumor inhibition effects. However, current data may not be sufficient to support this claim. It seems more likely that the addition of IFN α + IL-12 enhanced the generation of TA-specific CD8 T cells. First, the low dose OVA group showed a minimal increase of the OVA-specific CD8 T cells as compared to the control group $P = 0.0476$ (Figure 1E), indicating OVA alone can only slightly induce TA-specific CD8 T cells. Second, for the combo cytokine treatment without tumor antigen (IFN α + IL-12), the therapeutic effect is also superior (Figure 5A), and there is no difference for all aspects in terms of cytokine levels, T cell activation and, more importantly, tumor volume (Figure 5A, S5). The level of TEX T cells is even lower in the combo group (Figure 5C). These results demonstrate that the two cytokines may be the key to induce tumor-reactive T cells, although the authors found no antigen-specific CD8 T cells (Figure 5E). It seems that TA (*e.g.*, Trp2)-specific CD8T cells had little contribution to the anti-tumor efficacy.

We thank the Reviewer for this observation. We agree that the role of cytokines in enhancing tumor-reactive T cells is crucial, and we appreciate the Reviewer's insights into this point.

Regarding numbers of OVA specific CD8⁺ T cells in the liOVA group (previously referred as low dose group), we would like to clarify that we are measuring fraction of tumor reactive T cells out of all CD45⁺ cells. In the liOVA group we observe that out of all hematopoietic cells in the liver 1 to 2 % are OVA specific. When measured as a percentage of CD8⁺ T cells, OVA specific CD8 T cells represent on average 10% of all CD8 T cells in the mid dose group (previously referred as low dose) (**Fig. R2A**). These data support an expansion of OVA reactive T cells in the liver of the mice upon liOVA treatment.

We recognize that the cytokine combination (Ifna.II12) showed significant antitumor effect, as expected by the pro-inflammatory function of these two cytokines (**Fig. 5, A**). However, addition of TA (*i.e.*, Trp2) to the cytokine combo (Trp2.Combo) produced notable effects, especially in the early stages of tumor growth. The impact was particularly evident at the first time point (day 13), where we observed a delay in tumor growth that was statistically significant in the Trp2.Combo group when compared to cytokines alone (**Fig. R2B**). Moreover, it is important to highlight that only in the Trp2.Combo treated cohort we observed a mouse which completely eradicated B16-F10 LM. This point further supports the introduction of TA in combination with IL-12 and IFN α to trigger the most effective therapeutic response.

Fig.R2. A. Delivery of liver macrophage targeting Control LV, liOVA LV (mid or high dose) to mice before subcutaneous tumor implantation (Control LV 10^8 TU/mouse, mid dose 10^7 TU/mouse, high dose 10^8 TU/mouse). Liver OVA specific CD8⁺ T cells expressed as percentage of CD8⁺ T cells. Statistical analysis by Kurskal-Wallis test ($n=7$ mice/group). **B.** Tumor size measured by MRI. Direct comparison between mice treated with Ifna.II12 and with Trp2.Combo. Statistical analysis by Mann-Whitney test ($n=8, 9$ mice/group, for Ifna.II12 or Trp2.Combo treated mice, respectively).

Regarding the higher levels of TEX cells observed in the Trp2.Combo group, we think that this may be associated with the enhanced expansion of T cells in response to the TA delivery, as we also observed with the MC38.OVA and B16-F10 models when treating with liOVA (**Fig. 1,K**), Trp2.Combo (**Fig. 7, G**) or OVA.Combo (**Fig. 7,K**). In parallel to the increase fraction of TEX T cells, we found increased PEX T cells, suggesting that overall, addition of the TA promotes the expansion of a pool of PEX T cells.

The rationale for employing TAs in combination with cytokines is based on the premise that, while IFN α and IL-12 can shape the phenotype of activated T cell clonotypes, TAs have the ability to activate and expand naïve T cells expressing rare TCR clonotypes. Of note, animal models do not fully replicate the immune complexity of human tumors, making it challenging to precisely assess the contribution of TA delivery in combination with cytokines or other immune-activating treatments. Mouse tumor models typically develop over the course

of a few weeks, cancer cells are cultured *in vitro*, and accumulate immunogenic mutations without the evolutionary pressure exerted by the host immune system. As a result, these models usually express highly immunogenic TAs, that spontaneously activate TA specific T cells. To address this challenge, we utilized the melanoma LM expressing TRP2, a protein normally expressed by melanocytes. Due to thymic negative selection, TCR clonotypes recognizing TRP2 are largely counter-selected, resulting in a limited pool of TRP2-specific T cells. Interestingly, we found that the addition of TRP2 TA enabled the generation of TRP2-specific CD8⁺ T cells when combined with IL-12 and IFN α , as shown in **Fig. 5E**. In contrast, cytokines alone failed to expand TRP2-specific CD8⁺ T cells. Expansion of TRP2-specific CD8⁺ T cells was associated with improved tumor control. Overall, these findings support the rationale for combining TAs with the delivery of potent immune-activating cytokines, such as IL-12 and IFN α , to enhance anti-tumor immunity, especially for rare T cell clonotypes.

Minor:

5. Flow gating strategy needs to be provided

The gating strategy has now been included (**Supplementary fig. 1**) in the Materials and Methods section.

6. The variability of tumor growth for the control group in Figure 7E is large. A larger n number may be needed.

We acknowledge that the variability of tumor growth in the control group in **Fig. 7,E** is high. This variability reflects the inherent heterogeneity of tumor growth in this particular model (B16-F10) when implanted intrahepatically, as similarly observed in **Fig. 5,A**. Even if the variability is high, all the control mice developed tumors, and most of these tumors were bigger than the tumors in the treatment groups. Moreover, the aim of this particular experiment was to evaluate if addition of anti-PD1 could promote the expansion of PEX T cells in mice treated with Trp2.Combo. While increasing the number of mice in the control group could potentially reduce variability and improve the statistical power of the analysis, we believe that repeating the experiment is unlikely to provide significant additional insights or substantially alter the conclusions of the manuscript. The observed variability is inherent to this tumor model, and the current data already adequately support our key findings.

7. In some cases, ' α ' was written as 'a' in the manuscript.

We appreciate the Reviewer bringing this to our attention. In the manuscript, we have consistently used "IFN α " to refer to the cytokine as a protein, while "Ifna" was used as part of the experimental group names (e.g., Ifna.II12 or OVA.Ifna) to indicate mice injected with lentiviral vectors carrying the *Ifna* gene. We have carefully reviewed the manuscript to ensure consistent and accurate use of these terms, and any inconsistencies have been corrected.

Reviewer #2

This study presented a new method for gene therapy for the treatment of liver metastasis. The technique employs a lentiviral vector (LV) to ensure the coordinated expression of tumor antigens (TAs), IFN α , and IL-12 in the liver and tumor-associated macrophages (TAMs) through a single intravenous administration. The LVs incorporate a mannose receptor C type-1 (Mrc1) promoter, which is expressed mainly in resident macrophages, and microRNA target sequences that prevent transgene off-target expression in other cell types in the liver. The activated liver macrophages that facilitate the concurrent expression of TA and cytokines have proven to be effective in inhibiting tumor growth in different cancer models that form liver metastases. The majority of the mice showed complete response to the therapy. The inhibition

of the tumors is linked to a wide-ranging reprogramming of the tumor microenvironment (TME) as well as the activation of TA reactive CD8+ T cells. In addition to presenting a promising therapeutic tool, this research provides valuable insights into the mechanisms that promote immune activation and reprogramming in the metastatic liver cancer.

Major concerns:

1. The methodology of lentiviral vector (LV) delivery could be more clearly described in the study. It would be helpful to provide a detailed description of the LV administration process, including the dose, timing, and route of administration. This information would allow other researchers to replicate the study and also help in the development of future studies.

We thank the Reviewer for highlighting this important point. We recognize the significance of clearly describing our methods to enhance reproducibility. We have reported most of the information in the Figure legends and in the Material and Methods section in our initial submission, we have now revised both the main text and the methods section to provide a more thorough explanation to ensure that our findings can be replicated.

In brief, LVs were diluted in PBS to obtain the desired TU/mouse in a volume of 200 μ L. Before injection, mice were warmed under an infrared/red-light lamp and the LVs were delivered in the tail vein. For the injection of multiple LVs (e.g OVA.Combo, Trp2.Combo, TA33.Combo), the different LV preparations were mixed and diluted in PBS to achieve the desired TU/mouse. In prophylactic experiments, LVs were injected 14 days prior to tumor challenge, while in therapeutic experiments, LVs were injected after 3, 5 or 7 days post tumor placement. In general, LVs carrying the TA or IL-12, were injected at a dose of 10^7 TU/mouse while the LV carrying IFN α was administered at 10^8 TU/mouse. Specific timing and dosage of the LV injection are indicated either in the main text, in the figures or in the figure legends.

2. The study mainly used an artificial metastasis model by implanting colorectal and melanoma cells in the liver of mice. A natural metastasis model could have been used to better reflect the process of liver metastasis. The use of a more natural model would provide a more accurate representation of the effectiveness of the therapy in treating liver metastasis. This current model only demonstrates the effects of the LV therapies after metastatic seeding at the site but not the entire process.

In our study, we utilized three different liver metastasis (LM) models (MC38.OVA, B16-F10, and AKTPF) to account for variations in tumor origin, immunogenicity, and antigen expression. However, we recognize that a more natural model that better reflects tumor formation and metastasis process as observed in patients could have provided a more accurate representation of the effectiveness of the therapy. However, genetic models of LM are challenging to implement; they often exhibit high inherent variability, and only a small fraction of mice typically develop tumors. Additionally, such models or models where transplantable primary tumors give rise to spontaneous metastases would require the removal of the primary tumor, identification of novel tumor antigens, and time for liver tumor masses to grow, all of which present significant challenges and require surgical procedures that unfortunately at the moment we are not able to perform.

To address some of these limitations while still recapitulating key features of metastasis process, we conducted a new *in vivo* experiment, now included in the manuscript. To this aim, we first treated the mice with our lentiviral vectors (*i.e.*, TA33.Combo) and then introduced cancer cells into the splenic circulation to enable spontaneous liver metastatic seeding, mirroring the situation of patients who have had their primary tumors removed but still have

circulating cancer cells. As a result of this experimental design, we were able to show that treatment with TA33.Combo effectively protected the mice from metastatic dissemination in the liver (**Supplementary fig.10, A to C**). Notably, all the mice treated with TA33.Combo survived long term and were tumor free at the established endpoint at day 115 after LV treatment. However, please note that from a clinical perspective, we believe that the most appropriate first-in-human translation of our strategy would be in hepatic metastatic recurrences of previously resected CRC. If proven safe and successful, further clinical testing could expand to the neoadjuvant setting.

3. The study used a highly immunogenic system (OVA) for the analysis of tumor antigen (TA) expression. To make the study more applicable to natural metastasis models, the use of more natural antigens, such as the PMEL-T cell system, should be considered. This would provide a more accurate representation of the effectiveness of the therapy in treating liver metastasis.

We appreciate the Reviewer for highlighting the importance of considering immunogenicity and the applicability of natural antigens in our study. While we recognize the advantage of using a system like the PMEL-T cell model, we believe that incorporating it may not significantly modify the interpretation of our findings.

In our study, we employed three different LM models, two that were injected intrahepatically (*i.e.*, MC38.OVA, B16-F10) and one that was injected intrasplenically (*i.e.*, AKTPF). These distinct models address variations in tumor origin, immunogenicity, and antigen expression. Notably, in the B16-F10 model, we utilized the tumor-associated antigen Trp2, which is naturally expressed in melanocytes. In the Trp2.Combo-treated cohort, we successfully generated Trp2-specific CD8⁺ T cells, demonstrating the potential for our approach to elicit immune responses against naturally occurring tumor antigens. Additionally, as previously mentioned, we utilized the AKTPF colorectal cancer cells, which were originally derived from APC^{D716}; Kras^{G12D}; Tgfbr2^{-/-}; Trp53^{R270H}; Fbxw7^{-/-} mice. Through whole exome sequencing (WES) and RNA sequencing, we identified 33 putative tumor-specific antigens, which we delivered in the context of the TA33.Combo treatment (**Fig. 5, F-M**). Also in this setting, our treatment effectively induced the generation of CD8⁺ T cells capable of recognizing the delivered tumor antigens (**Fig. 5, L and M**).

Given the comprehensive nature of the models employed in the current study, we believe that adding another model like PMEL may not significantly modify the overall conclusions, as the employed models already provide valuable insights into the effectiveness of our therapeutic approach in the context of LM.

4. The study lacks a comprehensive analysis of the macrophages in the entire study. While the authors focused on the T cells, macrophages are the target cells that elicit the anti-tumor response. A more comprehensive analysis of the changes in macrophages under the influence of the LV system could be validated by scRNA sequencing followed by flow cytometry studies. This would provide a better understanding of the therapy's impact on the macrophages' properties and how they contribute to the anti-tumor response. Interaction strength between different cell types can also be evaluated.

We thank the Reviewer for these insightful comments regarding the analysis of macrophages in our study. We appreciate the emphasis on the role of macrophages in eliciting the anti-tumor response, and we recognize the need for a comprehensive understanding of their phenotype after treatment.

We have now expanded our single cell RNA sequencing analysis to better characterize macrophage subsets (**Supplementary fig. 5C**). We found that simultaneous co-delivery of cytokines and TA did not significantly alter the relative abundance of macrophage subsets in the tumor microenvironment (TME) as compared to TA only (**Supplementary fig. 5D**). This observation may be associated to the presence of IIOVA in the control group, which *per se* could drive expansion of inflammatory TAM clusters indirectly through CD8 T cell activation. However, LV-based expression of IL-12 or IFN α alone, and to a higher extent their combination, significantly modified the phenotype of TAMs, by increasing the expression of genes associated with antigen presentation (both MHC-I and MHC-II) and cytokine stimulation, while reducing the expression of pro-tumoral genes in most of the subsets we defined (**Supplementary fig. 5E**). Furthermore, we validated these findings on tumor myeloid cells, including tumor-associated macrophages, by employing multicolor flow cytometry. This analysis confirmed enhanced expression of MHC-II molecules, co-stimulatory receptors as well as other markers of immune activation (**Supplementary fig. 5, F to K**). We believe that these new analyses strengthen and expand our previous results, highlighting a general increase in the antigen presentation capabilities and pro-inflammatory state of liver and tumor macrophages of mice treated with our LV combination.

Reviewer #3

In the manuscript "*In vivo* armed macrophages curb liver metastasis through tumor reactive T cell rejuvenation" from Notaro et al, the authors used mouse models combined with single cell RNA sequencing to obtain important insights on the role of the immune system in liver metastasis. My comments are aimed specifically at the single cell RNA sequencing parts of the manuscript:

1) "To better investigate the effect of simultaneous IFN α and IL-12 delivery on immune cells, we performed single cell RNA sequencing (scRNA-seq) on CD45⁺ cells isolated from healthy liver parenchyma and matched LMs of treated mice. "

- Can you elaborate on the relatively high number (at least visually) of non CD45⁺ cells identified? Can you display somewhere in the text or supplementary which % they indicate?

We thank the Reviewer for this insightful comment. We appreciate the opportunity to clarify this point further.

Although we sorted for CD45⁺ cells, we observed a fraction of CD45⁻ cells in the single-cell RNA sequencing analysis. These cells accounted for 13.7 % of the total cells in the liver from the MC38.OVA experiment and 10.6 % of the total in the liver from AKTPF experiment. The percentage was much lower when sorting from tumor tissue (2.5% and 7.71% from MC38 and AKTPF LM respectively) (**Fig R3, A**). The higher percentage of CD45⁻ cells observed when sorting from hepatic tissue is likely due to the inclusion of non-immune cells, such as hepatocytes and endothelial cells, which might have been inadvertently sorted, possibly due to the autofluorescence commonly seen in liver tissue.

To better report this information, we have now included a pie chart that indicates the exact percentages of CD45⁺ and CD45⁻ cells for each sorted tissue, providing a clearer representation of the data (**Supplementary fig. 4A**). This addition should help visualize the proportion of non-immune cells in the dataset.

Although we initially expected a smaller fraction of non-hematopoietic cells, the presence of these populations in our dataset is unlikely to significantly impact the phenotypic

characterization of the hematopoietic cells analyzed in this study. Furthermore, for analyses focused on specific cell types, we subset and renormalized the data to better defined cell types and activation states as well as enable a more detailed and granular investigation.

Fig.R3, A. Pie chart indicating the percentage of CD45+ and CD45- cell types in the distinct single cell datasets.

2) “All cells were manually annotated based on their transcriptomic profile”
 - Can you include plots of known markers for all clusters? This makes the reading a bit easier instead of checking the supplementary tables.

We thank the Reviewer for this helpful suggestion. We have now included plots showing the expression of top markers as well as key well-known differentially expressed genes that define cell identity for all cell clusters (**Supplementary fig. 4, D and E**). We believe this will improve the readability of the manuscript and the interpretation of the cell types indicated in the plots without needing to consult the supplementary tables.

3) “Analyzing together all APC cell types, we observed in both liver and LM that OVA.Ifna increased the expression of genes involved in MHC-I antigen presentation, such as MHC-I subunits (H2-Q7, 207 H2-T22, H2-Q4, H2-K1, H2-M3, H2-D1) and peptide transporters (Tap1, Tap2), and of interferon 208 stimulated genes (Ifi44, Irf7, Isg20, Oas1a, Oas1g) compared to the liOVA group (Fig. 2, E and 209 F).”

- For sake of clarity, were these genes identified exactly how? By comparing different clusters to each other? Or all cells from specific group versus each other? Can you elaborate on how they were identified?

- Please harmonize cell type nomenclature. In fig 2G you refer to granulocytes, which I assume are the neutrophils in 2C and 2D. If DCs were combined, please clarify as well.

We thank the Reviewer for this thoughtful feedback. When first analyzing our data, we investigated which biological processes were modulated by our treatment in monocytes, macrophages and KCs (**Supplementary Fig. 5A**). Building on what we observed with this analysis we then manually selected well-known genes involved in these biological processes. The expression of the single genes was then measured in all APCs identified in the study. For example, in Figure 2E we are plotting the expression level (*i.e.* fraction of cells and scaled average expression) of selected genes expressed by all APCs in all the distinct groups. P values were calculated by comparing the expression between each treatment group and the

liOVA group. The methodology has been clarified in the figure legend of Figure 2 and in the Material and Methods section.

Regarding the nomenclature issue, we have now included a figure that displays all distinct cell types separately, without combining them, to provide a clearer depiction of the populations analyzed (**Fig. 2G and Supplementary Fig.4F**).

4) “expanded TCR clonotypes were mostly shared between the liver and LM. Of note, CD8+ T cells bearing expanded and shared TCR clonotypes clustered with CD8+ T effector cells in the UMAP plot, representing putative treatment-induced tumor specific CD8+ T cells (Fig. 6B)” - The authors show an interesting enrichment of enriched clonotypes in very well-defined cell clusters. However, as shown by Sundell et al 2022 DOI: 10.1093/bfpg/elac044, sometimes TCR related genes are captured by both 3' and 5' single cell maseq protocols. These genes can eventually lead to biased cluster identification, which in the context of intense clonotype expansion can be particularly harmful. I would recommend the authors reanalyse the data after exclusion of TCR related genes.

We sincerely thank the Reviewer for this valuable feedback. To address this point, we thoroughly tested whether TCR genes could introduce bias in cluster identification within our dataset.

First, we checked whether TCR genes were among the top 30 genes by average fold change in each defined cluster. We identified around 500 genes (i.e. 30 genes times the number of clusters, minus common genes), and between these, we discovered 21 (4.2%) and 7 (1.2%) TCR genes in the MC38.OVA and AKPTF experiments respectively. We observed an enrichment of 11 TCR genes in the top 30 markers of CD8 Teff3 in the MC38.OVA experiment but not in the AKPTF experiment which contained only one TCR gene in the top 30 markers of CD8 Teff3. This observation is in line with the possibility that TCR genes could impact on cluster identification.

Building on this observation, we tested to what extent TCR genes impacted the UMAP projection and clustering analysis. We removed 254 TCR genes from our dataset as described in ². We repeated our entire analysis including clustering and cell type annotation. We carefully compared the results obtained with and without TCR genes. Our analysis revealed that the removal of TCR genes did not significantly impact the unbiased clustering and cell type composition (**Supplementary fig.13, A and B**). The new clusters and cell types contained an average of 87% of the same cells as the original clusters and cell types, with the new CD8 Teff3 cells comprising 93% or 85% of the original CD8 Teff3 cells in the MC38.OVA or AKTPF experiment respectively. The only population that was substantially affected by this new analysis was the one identified initially as early CD8 T cells. However, these cells, which are transcriptionally similar to Naïve T cells by definition fall predominantly with the new clustering in the Naïve CD8 T cell cluster in the MC38 cohort (**Supplementary table R1**). Of note, we found that a high proportion (0.24; n = 52 cells) from the cells initially annotated as early CD8 T cells in the AKTPF cohort fall within the NK cluster. We believe this may be due to the removal of TCR genes, which could lead to these cells being mistakenly annotated as NK cells. Importantly, tumor antigen (TA)-specific T cells, characterized by expanded TCR clonotypes, remained enriched in the CD8 Teff clusters regardless of whether TCR genes were included or removed, in both the MC38.OVA and AKPTF cohorts (**Supplementary fig.13, C and D**).

In summary, while we acknowledge the importance of removing TCR genes when clustering single-cell data, we agree with the Reviewer that this approach may be relevant for certain studies but is not necessarily applicable to all. Based on our findings when comparing

the two methodologies, we conclude that, for our study, the exclusion of TCR genes does not substantially affect clustering and cell type identification. We have now included this analysis in the Results section.

5) "Base call files obtained as result from the Illumina sequencing were converted into FASTQ files, aligned to the mouse reference genome, and quantified with the 10x Genomics Cell Ranger Software (v7.2.0) using default parameters."

- Which genome version was used?

We appreciate the Reviewer for the diligence in reviewing the manuscript, and we are sorry for missing this important information for the reproducibility of the study. The reference genome used was GRCm38, and this information is now included in the materials and method of the manuscript.

Reviewer 5

Notaro et al describe a *Mrc1*-driven LV -mediated gene therapy approach to increase immunity against liver metastases (LM). They use a variety of LM models, expressing either a surrogate TA, a well characterized natural TA, or even a model in which 33 putatively immunogenic peptides were discovered. In all cases, the combined LV-mediated expression of CD74-conjugated TA, IFN α and IL12 lead to a superior anti-tumor T-cell immunity in the LM environment, which is demonstrated via the assessment of LM development, via flow cytometry and scRNAseq of the immune compartment and spatial transcriptomics, the latter showing an enhanced colocalization of APCs and T cells. Finally, LV TA/IFN α /IL12 treatment increased the effect of anti-PD1 treatment in models that are otherwise insensitive to anti-PD1. This is a strong story with a wealth of data. However, some important remarks remain:

1) The authors assume that any effects of the LV-OVA or other LV-mediated treatment is due to liver macrophages, because gene expression is *Mrc1*-driven. This is a huge assumption, that is not backed up by any proof. Since the LVs are injected iv, other *Mrc1*+ macrophages (*Mrc1* is of course not restricted to liver macrophages), eg in lymphoid organs, may play an important role. A thorough analysis of which macrophage populations in the body, and certainly in the lymphoid organs, take up the LV and present OVA or other TA is crucial.

We thank the Reviewer for raising this important point. We apologize for not clearly referencing our prior work investigating the biodistribution of the LV platform ¹. In our previous work, we extensively characterized the biodistribution of LVs upon systemic delivery to immunocompetent mice as well as the expression pattern of a transgene under the transcriptional control of the mannose receptor c-type 1 (*Mrc1*) promoter and post-transcriptional control of microRNA (miRNA) target sites for miR-122-5p and miR-126-3p. Previously, we showed that the transgene expression obtained by employing this LV-based platform is selective for liver macrophages, with negligible expression in splenic MRC1+ macrophages. Of note, we showed no expression in other biological compartments such as blood, bone marrow, lung, sub-iliac lymph nodes, small intestine, or brain. In our previous work, we also analyzed transgene expression in livers bearing colorectal cancer (CRC) metastases and found that cancer cells were virtually untransduced. This information is now more clearly referenced in the revised manuscript.

To address the potential effects of antigen presentation/cytokine production deriving from off-target transduction of splenic macrophages,

We concur with the referee that even low expression of antigens of these cytokines in other organs may be to some extent drive the therapeutic response observed in our study. As mentioned earlier, we have detected expression of transgene upon LV delivery in Mrc1⁺ macrophages in the spleen, but not in lymphoid organs in our previous study¹. Therefore, to investigate to what extent expression of cytokines and TA in the spleen were involved in driving anti-tumoral immune responses, we performed a new experiment where we splenectomized the mice before tumor placement and OVA.Combo treatment. Even in splenectomized mice, OVA.Combo treatment impaired tumor growth. Moreover, OVA.Combo enabled the expansion of OVA specific CD8⁺ T cells in circulation and in the liver of mice compared to controls (**Supplementary fig.3, H to J**). Expanded OVA specific CD8⁺ T cells displayed a phenotype consistent with earlier experimental results in unsplenectomized mice. These new data suggest that the therapeutic effects of the simultaneous delivery of TA and cytokines are independent from the off-target expression of the transgenes in splenic macrophages.

2) Another important issue is that the models of liver metastasis used in this study are not really reflecting liver metastasis. The authors always employ xenotopically implanted tumors, directly injected in the liver. It would be important to show that some of the crucial LV-mediated treatments also work in a model of true liver metastasis by CRC or melanoma cells, i.e. models whereby cancer cells arrive in the liver and seed the liver parenchyma to establish LM. For example, cancer cells can be injected in the spleen and allowed to seed the liver. Models of spontaneous LM formation, starting from a primary tumor, would even be better, although not many such models are available.

As mentioned by the Reviewer in the summary of our study we utilized three different liver metastasis (LM) models (MC38.OVA, B16-F10, and AKTPF) to account for variations in tumor origin, immunogenicity, and antigen expression. However, we recognize that a more natural model that better reflects tumor formation and metastasis process as observed in patients could have provided a more accurate representation of the effectiveness of the therapy. However, genetic models of LM are challenging to implement; they often exhibit high inherent variability, and only a small fraction of mice typically develop tumors. Additionally, such models or models where transplantable primary tumors give rise to spontaneous metastases would require the removal of the primary tumor, identification of novel tumor antigens, and time for liver tumor masses to grow, all of which present significant challenges and require surgical procedures that unfortunately at the moment we are not able to perform. However, to account for natural dissemination of cancer cells in the liver of the mice, we performed intrasplenic injections when challenging the mice with the AKTPF tumor model. This information is now included in the description of the experiments in the Results section.

Moreover, to better recapitulate key features of metastasis process, we conducted a new *in vivo* experiment, now included in the manuscript. We first treated the mice with TA33.Combo and then introduced cancer cells into the splenic circulation to enable spontaneous liver metastatic seeding, mirroring the situation of patients who have had their primary tumors removed but still have circulating cancer cells. As a result of this experimental design, we were able to show that treatment with TA33.Combo effectively protected the mice from metastatic dissemination in the liver (**Supplementary fig.10, A to C**). Notably, all the mice treated with TA33.Combo survived long term and were tumor free at the established endpoint at day 115 after LV treatment. However, please note that from a clinical perspective, we believe that the most appropriate first-in-human translation of our strategy would be in hepatic metastatic recurrences of previously resected CRC. If proven safe and successful, further clinical testing could expand to the neoadjuvant setting.

3) Lines 152-154. Reduced exhaustion of CD8+ T cells is assumed based on a lower MFI of only one marker, PD1. One marker expression is not sufficient to draw such a conclusion.

We thank the Reviewer for raising this point and we agree that PD1 expression might not be sufficient to assume reduced exhaustion of this cell population. We have now revised the main text to highlight the increase Ly6c expression on these cells, which, in combination with the reduced expression of PD1, might indicate an increased activation and not necessarily reduced exhaustion (**Supplementary Fig. 2L**)

4) Overall, most of the analyses in flow cytometry and scRNAseq are rather centered on the lymphocyte compartment, while there is no real proof delivered that CD8+ T cells are the ultimate anti-tumor effector cells. More granular information on the myeloid compartment is equally informative, as it cannot be excluded that these cells also play a role as effector cells.

For example, the UMAP plots shown in Fig 2C-D are not optimal, in the sense that the analysis could be improved. Monocytes: which ones, as there are subsets? Macrophages is a very broad term and no annotations of macrophage subsets are given. Mo-DC are likely cDC2....

Likewise, Fig 2G can provide more granular information. I do get that the authors want to show a general trend upon LV-cytokine treatment, but they have the single-cell data to discriminate between different DC subsets (cDC1, cDC2, pDC,...), between different granulocytes (neutrophils, eosinophils,...), likely also different macrophage subsets...

We thank the Reviewer for the insightful comment. We have addressed the relevance of CD8+ T cells in our study by performing T cell depletion experiments, which highlighted their critical role in the anti-tumor response (**Fig. 4M**). However, we agree with the referee that activated CD8 T cells may play an indirect role in the observed therapeutic effect. For example, CD8 T cells may promote the function of other cell types, such as macrophages, that may be the ultimate effectors of the therapeutic activity observed upon our intervention. We think that the Reviewer will concur with us that addressing this, for example through macrophage depletion studies may possess technical limits for us, since cytokines are produced selectively by liver macrophages upon transduction with our LV-based platform. Nevertheless, better addressing genetic changes in macrophages in our single cell RNA sequencing analysis may at least in part, shed light on their role in our study.

To this aim, we have now expanded our single cell RNA sequencing analysis to better characterize macrophage subsets (**Supplementary fig. 5C**). We found that simultaneous co-delivery of cytokines and TA did not significantly alter the relative abundance of macrophage subsets in the tumor microenvironment (TME) as compared to TA only (**Supplementary fig. 5D**). This observation may be associated to the presence of IIOVA in the control group, which *per se* could drive expansion of inflammatory TAM clusters indirectly through CD8 T cell activation. However, LV-based expression of IL-12 or IFN α alone, and to a higher extent their combination, significantly modified the phenotype of TAMs, by increasing the expression of genes associated with antigen presentation (both MHC-I and MHC-II) and cytokine stimulation, while reducing the expression of pro-tumoral genes in most of the subsets we defined (**Supplementary fig. 5E**). Furthermore, we validated these findings on tumor myeloid cells, including tumor-associated macrophages, by employing multicolor flow cytometry. This analysis confirmed enhanced expression of MHC-II molecules, co-stimulatory receptors as well as other markers of immune activation (**Supplementary fig. 5, F to K**). We believe that these new analyses strengthen and expand our previous results, highlighting a general increase in the antigen presentation capabilities and pro-inflammatory state of liver and tumor macrophages of mice treated with our LV combination. Although, this analysis does not

address their role as effector cells, it highlights their contribution in antigen presentation and T cell activation specially in the tumor.

We have now changed the nomenclature of moDCs in cDC2, as the Reviewer corrected pointed out, at a more refined analysis the gene expression profile of these cells resembles that of cDC2. These cells indeed are characterized by higher expression *Irf4* compared to other clusters and lower expression of *Cd14*. Initially, we classified these cells as moDCs due to their resemblance to moDCs identified in our previous work ¹. We thanks the Reviewer for this indication.

We have modified Figure 2G as requested by the Reviewer, as well as Supplementary Figures 4F, 7M and 11D to distinguish between the distinct subsets of DCs and granulocytes.

Overall, these new analyses have strengthen our previous results, highlighting a general increase in the antigen presentation capabilities and pro-inflammatory state of liver and tumor macrophages of mice treated with OVA.Combo.

5)Fig 2A-B: Explain better what we see in the figure. It is metastatic nodules (in B) versus healthy liver tissue from the same livers?

In Figures 2A and B we reported the UMAP projection of cells isolated from the healthy liver lobule and from LM obtained from matched animals. We have now better described the figure in the Results section and in the figure legends.

REFERENCES

1. Kerzel T, *et al.* In vivo macrophage engineering reshapes the tumor microenvironment leading to eradication of liver metastases. *Cancer cell* **41**, 1892-1910 e1810 (2023).
2. Sundell T, Grimstad K, Camponeschi A, Tilevik A, Gjertsson I, Martensson IL. Single-cell RNA sequencing analyses: interference by the genes that encode the B-cell and T-cell receptors. *Brief Funct Genomics* **22**, 263-273 (2022).
3. Car BD, *et al.* Role of interferon-gamma in interleukin 12-induced pathology in mice. *The American journal of pathology* **147**, 1693-1707 (1995).
4. Carretero-Iglesia L, *et al.* High Peptide Dose Vaccination Promotes the Early Selection of Tumor Antigen-Specific CD8 T-Cells of Enhanced Functional Competence. *Frontiers in immunology* **10**, 3016 (2019).
5. Fan T, Zhang M, Yang J, Zhu Z, Cao W, Dong C. Therapeutic cancer vaccines: advancements, challenges, and prospects. *Signal Transduct Target Ther* **8**, 450 (2023).
6. Utzschneider DT, *et al.* High antigen levels induce an exhausted phenotype in a chronic infection without impairing T cell expansion and survival. *The Journal of experimental medicine* **213**, 1819-1834 (2016).
7. McLane LM, Abdel-Hakeem MS, Wherry EJ. CD8 T Cell Exhaustion During Chronic Viral Infection and Cancer. *Annual review of immunology* **37**, 457-495 (2019).

8. Zehn D, Thimme R, Lugli E, de Almeida GP, Oxenius A. 'Stem-like' precursors are the fount to sustain persistent CD8(+) T cell responses. *Nature immunology* **23**, 836-847 (2022).

Point-by-point response to Reviewers (Second round of revision)

We thank the Editor and the Reviewers for the positive feedback and insightful review of our manuscript.

Reviewer 1

The authors have adequately addressed my comments. This manuscript can be accepted for publication.

We thank the Reviewer for appreciating our work and for the insightful comments.

Reviewer 2

All questions addressed. Thank you for the revision.

We thank the Reviewer for appreciating our work and for the insightful comments.

Reviewer 4

We thank the Reviewer for appreciating our work and for the insightful comments.

Reviewer 5

The authors sufficiently addressed my concerns

We thank the Reviewer for appreciating our work and for the insightful comments.

Reviewer 6

The authors have addressed the comments raised by the previous Reviewer #3 regarding bioinformatic analysis. I have only one minor comment: The co-localization score heatmaps (supplementary Fig. 14 C and D) are somewhat unclear due to the absence of row and column labels, as well as a legend title. Adding these elements would improve readability.

We thank the Reviewer for appreciating our work and for helping us to clarify Supplementary Figure 14. We have improved Supplementary Figure 14 C and D as suggested by the Reviewer.